# Towards a Comprehensive Scaling Law of Mixture-of-Experts

## Abstract

Mixture-of-Experts (MoE) models have become the consensus approach for enabling parameter-efficient scaling and cost-effective deployment in large language models. However, existing scaling laws for dense models are inapplicable to MoE models, which stems from three critical challenges: the multiplicity of influencing factors, their intricate coupling relationships and the non-monotonic nature of their performance impacts. They collectively necessitate a fine-grained investigation into MoE-specific scaling laws. In this work, we perform a systematic decomposition of MoE settings, identifying five key factors that influence model performance from both size and structural perspectives (data size ($D$), total model size ($N$), activated model size ($N_a$), number of active experts ($G$) and the ratio of shared experts ($S$)). Specifically, we design $450$ controlled experiments to characterize their marginal effects, ultimately constructing a comprehensive and precise joint MoE scaling law that considers all essential factors. Furthermore, we derive the theoretically optimal and practically efficiency-aware optimal configurations for $G$, $S$ and $N_a/N$ with detailed analyses. Our results demonstrate that the optimal settings for $G$ and $S$ are independent of both the model architecture and data size. With the scaling of $N$, the optimal activation parameter ratio of $N_a/N$ becomes sparser. Our proposed MoE scaling law could function as an accurate and insightful guidance to facilitate future MoE model design and training.

## 1 Introduction

Large language models (LLMs) have been widely verified and utilized in our daily lives. It is impressive and lucky to discover that LLMs can continuously expand its ability boundaries with increasing model and training data sizes. The scaling laws of LLMs (Kaplan et al., 2020; Hoffmann et al., 2022; Sun et al., 2025), which could predict the model loss based on crucial factors (e.g., data/model sizes) before training, shed lights on the promising way of wisely selecting appropriate model structures and settings before experiments and continuously enhancing the ability of LLMs under given training budget or environment constraints. Recently, **Mixture-of-Experts (MoE)** becomes one of the mainstream structures broadly used in powerful industry-level LLMs (Dubey et al., 2024; Liu et al., 2024; Sun et al., 2024; Liu et al., 2025; Qwen Team et al., 2025; OpenAI et al., 2025). Different from the original dense architecture that involves all parameters in the forward process, MoE often adopts multiple experts (e.g., FFNs) with a router to automatically select which experts should be activated for the current token. The sparse activation of experts in MoE could largely benefit from increasing total model sizes while maintaining efficient model inference.

With the thriving in efficient LLMs, lots of efforts have been dedicated to MoE architectures. The shared expert is proposed to capture general knowledge robustly (Dai et al., 2024; Sun et al., 2024). We also notice the trend of increasing (activated and total) expert numbers (Kimi et al., 2025; Liu et al., 2024; OpenAI et al., 2025). In this case, existing scaling laws of either dense models (Kaplan et al., 2020; Hoffmann et al., 2022) or MoE models (Krajewski et al., 2024; Wang et al., 2024b) cannot perfectly predict the model performance under the updated popular MoE structures and settings. The community urgently requires a new MoE scaling law to accurately guide model training.

To comprehensively explore the central factors of MoE models that largely impact the model performance, we first take the classical factors of **data size** ($D$) and **total model size** ($N$) also marked in dense scaling laws into consideration. Besides, the **activated model size** ($N_a$) functioning in the

forward process is essential in MoE models. For the expert aspect, we note the **number of activated experts** $G$ as another essential factor ($N_a$ and $G$ collaboratively determine the expert size). Moreover, the **ratio of shared experts in activated experts** ($S$) is also modeled ($S$ and $G$ collaboratively set the specific numbers of shared and routed experts). We attempt to build our scaling laws of MoE based on the above five factors. The challenges mainly locate in three aspects: (a) our MoE scaling law considers more comprehensive factors $D$, $N$, $N_a$, $G$, $S$ compared to existing scaling laws. (b) Our preliminary experiments imply that some factors have a non-monotonic impact on loss, which are more challenging to fit. (c) There exists mutual coupling relationships among these factors, which multiplies the challenges of constructing the final joint scaling law.

To accomplish our MoE scaling laws, we first select a reasonable and wider parameter range for the five essential factors $D$, $N$, $N_a$, $G$, $S$ and then conduct experiments 450 to record the corresponding MoE losses of different parameter settings. Based on these experimental results, we first decide the basic scaling law formation with the fundamental $D$, $N$ factors following Hoffmann et al. (2022). Next, we discover the marginal effects of $N_a$, $G$, $S$ respectively, whose impacts on model losses are surprisingly non-monotonic and are coupled to other factors. Finally, we obtain the joint MoE scaling laws formulated as follows:

$$L(N, D, N_a, G, S) = (eG + \frac{f}{G} + mS^2 + nS) * (\frac{1}{N^\alpha} + \frac{k}{N_a^\alpha} + h\frac{N_a}{N}) + \frac{a}{N^\alpha} + \frac{b}{D^\beta} + \frac{c}{N_a^\alpha} + \epsilon. \quad (1)$$

which could satisfactorily predict MoE models' losses with larger data/model sizes (e.g., up to 9B total model size and 100B trained tokens) and different MoE settings (e.g., up to 256 experts, 20 activated experts and 70% ratio of shared experts).

Based on our MoE scaling laws, we conduct in-depth analyses and discover the following implications: (a) *the optimal number of activated experts is around 7* for real-world classical MoE settings considering its effectiveness. (b) *Too dense/sparse MoE structures are not performance-optimal*. The $20\% \sim 43\%$ activated parameter ratios ($N_a/N$) are theoretical optimal for $N$ from 1T to 20B. Considering the cost, the practical efficiency-aware optimal ratios range from $5\% \sim 9\%$. (c) *The existence of shared expert is essential*, while the best ratio $S$ of shared experts to all activated experts ranges from $13\% \sim 31\%$ with marginal loss disturbances. We are confident that our MoE scaling laws with the above observations and insights could provide a more comprehensive understanding and more accurate performance prediction of MoE models with different settings, looking forward to facilitate LLM community in future MoE model design and training.

## 2 PRELIMINARY

### 2.1 MoE ARCHITECTURE

The Mixture-of-Experts (MoE) architecture modifies standard Transformer by replacing the dense Feed-Forward Networks (FFNs) with a set of independent experts, where each expert is usually an FFN of the same size (Fedus et al., 2022; Zhou et al., 2022; Jiang et al., 2024). Typically, for each token, the router selects and activates only a small subset of these experts. This design allows model size to grow by adding experts, while keeping the computational cost nearly unchanged. This makes it possible to scale models to very large sizes without a proportional increase in cost. Considering the trend of training-/inference- efficient LLM, MoE has become a mainstream and effective framework for building industry-level LLMs balancing model performance and computational efficiency.

### 2.2 EXISTING SCALING LAWS OF LLMs

**Scaling Laws of Dense LLM.** Scaling laws describe the relationship between key factors such as total model size $N$, data size $D$ and the loss $L$. Classical scaling laws include the Chinchilla scaling law (Hoffmann et al., 2022), which states that $L$ follows a power-law dependence on $N$ and $D$, written as $L(N, D) = a/N^\alpha + b/D^\beta + \epsilon$. It consists of three terms: the first and second terms capture the limitations imposed by finite model size and finite data size, respectively. The last term, $\epsilon$, represents the irreducible error that arises from the inherent uncertainty in the training data.

**Scaling Laws for MoE.** Unlike dense models, MoE introduces new structures with additional factors (e.g., the activated model size, the number of activated experts, the ratio of shared experts, etc),

whose effects on model loss are non-monotonic and often interdependent. Existing scaling laws for dense models are insufficient to guide MoE's model design, which motivates the development of new scaling laws tailored to MoE with these new-added factors. Recent studies have investigated MoE scaling laws based on certain MoE-specific factors, including the granularity of activated experts (Krajewski et al., 2024) and the activated model size (Ludziejewski et al., 2025). Different from them, our scaling law is more comprehensive and quantitatively defined considering five factors. As a result, our scaling law provides a more accurate fit to the loss, as shown in Fig. 5.

## 3  Experimental Setup

We systematically analyze the five key factors of MoE that influence training dynamics: data size $D$, total model size $N$, activated model size $N_a$, number of activated experts $G$ and ratio of shared experts in activated experts $S$. To isolate their individual effects, we conduct $450$ controlled experiments divided into several groups, each group varying a single target factor while holding the others fixed. This setup enables a clear assessment of how each factor impacts the validation loss.

Formally, we define the number of shared experts as $n_s$, the number of routed experts as $n_e$, the number of activated routed experts as $n_k$, the head dimension as $d_{\text{head}}$, the hidden dimension as $d_{\text{hidden}}$, the expert dimension as $d_{\text{expert}}$, the number of heads as $n_h$ and the number of layers as $l$. Based on these definitions, the following relationships hold:

$$G = n_k + n_s, \quad N_a \approx \big(4d_{\text{head}} \cdot n_h + 3Gd_{\text{expert}}\big)d_{\text{hidden}} \cdot l, \tag{2}$$

$$S = \frac{n_s}{G}, \quad N \approx \big(4d_{\text{head}} \cdot n_h + 3d_{\text{expert}}(SG + n_e)\big)d_{\text{hidden}} \cdot l. \tag{3}$$

An MoE layer consists of multiple experts and a router that assigns tokens, often using a Top-$K$ routing strategy with an auxiliary balance loss to ensure expert utilization. Training typically adopts standard optimizers such as AdamW (Kingma & Ba, 2014) with parallelism techniques (data, model and expert parallelism) for scalability. We select the widely-used classical MoE structures with architectural details provided in Appendix B. All experiments employ the Warmup-Stable-Decay (WSD) learning rate scheduler (Hu et al., 2024). For studies involving different values of $D$, we reused the same warmup and stable phases across runs to avoid redundant computation and reduce resource usage. All models are trained on a subset of the Dolma V1.7 dataset (Soldaini et al., 2024).

To fit the correlations between validation loss and five key factors, we systematically conduct experiments within controlled ranges of language model pretraining, with total model size $N \in [133\text{M}, 3.4\text{B}]$ and training data sizes $D \in [10\text{B}, 50\text{B}]$ tokens. We additionally vary the activated model size $N_a \in [30\text{M}, 2.2\text{B}]$, the number of activated experts $G \in [1, 20]$ and the ratio of shared experts $S \in [0.0, 0.7]$. For validation, we further extend to larger model and data sizes (up to 9B total model size and 100B trained tokens) and different MoE settings (up to 256 experts, 20 activated experts and 70% ratio of shared experts), successfully verifying the effectiveness and generalization ability of our scaling laws. The complete experimental settings are reported in Appendix L.

## 4  Our Scaling Laws for MoE

MoE has gradually emerged as a primary solution for the continuous scaling of model sizes and efficient deployment of LLMs. The core factors of MoE models exhibit higher complexity and strong inter-factor coupling compared to dense models and such inherent complexity renders the classical Chinchilla (Hoffmann et al., 2022) and OpenAI (Kaplan et al., 2020) scaling laws insufficient to guide the design of MoE architectures.

We aim to build a comprehensive and accurate MoE scaling law. In the following, we sequentially elaborate on the marginal effects of each core factor on MoE model performance via controlled-factor experiments, encompassing the total model size ($N$), data size ($D$), activated model size ($N_a$), number of activated experts ($G$) and the ratio of shared experts in activated experts ($S$). Building upon these analyses, we further derive the methodology underlying our joint MoE scaling law.

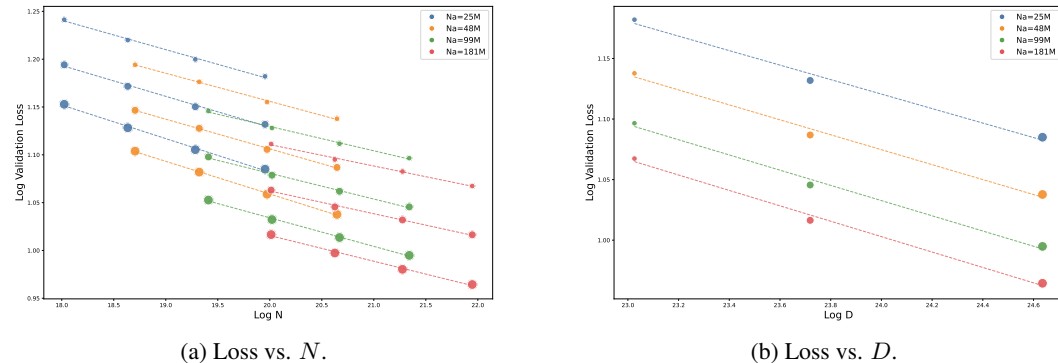

(a) Loss vs. $N$.

(b) Loss vs. $D$.

Figure 1: Marginal effects of validation loss with respect to N and D under the logarithmic coordinate system. Data point sizes are directly proportional to D.

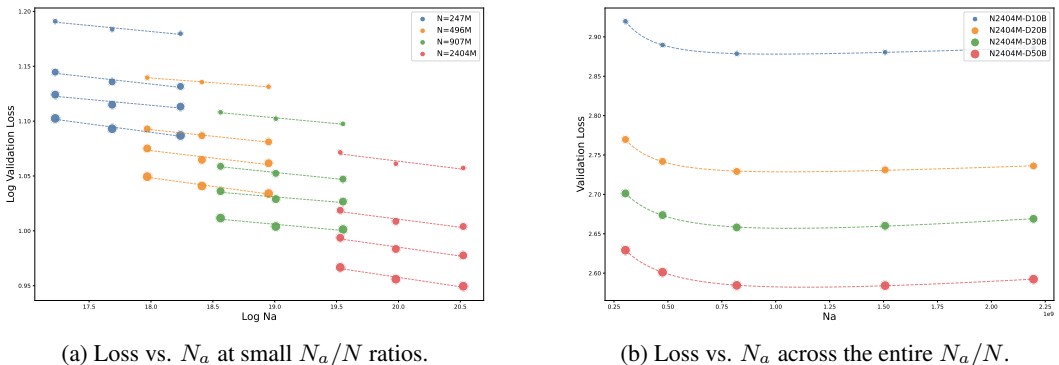

(a) Loss vs. $N_a$ at small $N_a/N$ ratios.

(b) Loss vs. $N_a$ across the entire $N_a/N$.

Figure 2: Marginal effects of validation loss with $N_a$. (a) illustrates the power-law-like marginal relationship between loss and $N_a$ under smaller $N_a/N$. (b) indicates that loss oscillates and increases as $N_a/N$ becomes increasingly large. Data point sizes are proportional to $D$.

## 4.1 THE BASIC MOE SCALING LAW'S FORM WITH $N$ AND $D$

Total model size $N$ and data size $D$ constitute two primary factors influencing the performance of LLMs. Leveraging the scaling laws of dense models as a foundation, we examine whether $N$ and $D$ in MoE still conform to a power-law relationship. As illustrated in Figure 1, a distinct power-law relationship is observed between validation loss, total model size $N$ and data size $D$.

Specifically, we increase $N$ with $G$, $S$, $D$ and $N_a$ unchanged. By performing logarithmic transformations on both the validation loss and $N$, a linear relationship is observed across different ranges of model sizes—this confirms a significant power-law relationship between the loss and $N$. Similarly, a significant power-law relationship also exists between the loss and $D$. Furthermore, in Figure 1, we find that experimental groups with the same model size but varying data sizes exhibit a translation along the $y$-axis, which implies that $N$ and $D$ are mutually independent. From these observations, we conclude that the loss for MoE with respect to the total model size $N$ and data size $D$ is as:

$$L(N, D) = \frac{a}{N^\alpha} + \frac{b}{D^\beta} + \epsilon, \tag{4}$$

which has the same form of the Chinchilla scaling law (Hoffmann et al., 2022). The specific fitting results of Eq. 4 are provided in Appendix C.2.

## 4.2 IMPACT OF ACTIVATED MODEL SIZE $N_a$

Activated model size $N_a$ is a critical factor specific to MoE architectures that governs the balance between the model performance and efficiency. To gain deeper insights into the scaling law with

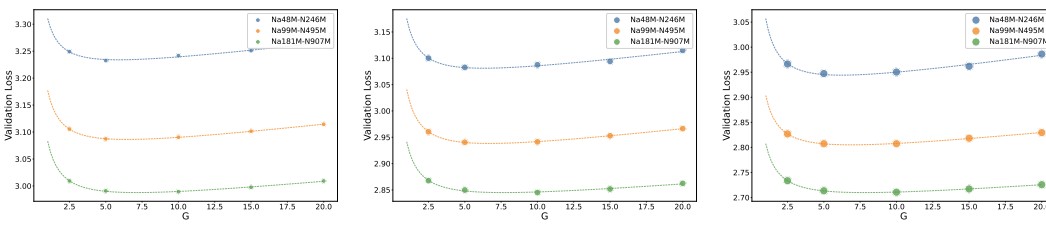

(a) Loss vs. $G$ with 10B data size.    (b) Loss vs. $G$ with 20B data size.    (c) Loss vs. $G$ with 50B data size.

Figure 3: Marginal effects of validation loss with respect to $G$. (a), (b) and (c) illustrate the marginal relationship between loss and $G$ under different $D$ and $N$. Data point sizes are proportional to $D$.

$N_a$ as an independent factor, we conducted multiple controlled experiments where $N_a$ was the only varying factor. The formula followed by the controlled variable of $N_a$ is detailed in Appendix H..

In Figure 2, the scaling of activated parameters $N_a$ is achieved by increasing the expert dimension, while the total model size $N$ is maintained constant through a corresponding reduction in the number of routed experts. When the ratio of $N_a$ to $N$ is small, a power-law-like relationship is exhibited between $N_a$ and validation loss. However, as this ratio increases, the validation loss demonstrates a tendency to rise gradually, leading to an overall distribution that resembles a hook-like function, which is formalized as follows:

$$L(N_a) = \frac{c}{N_a^\gamma} + hN_a + \iota. \tag{5}$$

Next, we proceed to integrate the relationship involving $N_a$ with those of data size $D$ and total model size $N$. We performed hyperparameter fitting under different configurations of $D$ and $N$, with the results presented in Figure 8. We observe that $\iota$ exhibits a negative correlation with both $D$ and $N$, following a power-law distribution. In contrast, $c$ and $\gamma$ exhibit oscillations with variations in $N$ and $D$, indicating that they bear no systematic relationship to $N$ and $D$. Furthermore, $h$ is negatively correlated with $N$, exhibiting an inversely proportional relationship, while showing no dependence on $D$. Therefore, the joint scaling law of $N$, $D$ and $N_a$ is concluded as follows:

$$L(N, D, N_a) = \frac{c}{N_a^\gamma} + h\frac{N_a}{N} + \iota L(N, D). \tag{6}$$

$L(N, D)$ denotes the basic scaling law sharing the same form in Eq. 4. Notably, our fitting results reveal that $\gamma \approx \alpha$. Hence, the final scaling law that governs $L(N, D, N_a)$ is formalized as follows:

$$L(N, D, N_a) = \frac{a}{N^\alpha} + \frac{b}{D^\beta} + \frac{c}{N_a^\alpha} + h\frac{N_a}{N} + \epsilon. \tag{7}$$

We explored alternative relationship forms of $L(N_a)$, performed relevant comparative experiments and derivations and ultimately established the aforementioned formulas describing the scaling law of $N$, $D$ and $N_a$. Eq. 7 implies that $N_a$ has an optimal value, which seems to be contrary to the prevalent assumption that "a larger $N_a$ yields lower loss". It is because that when $N_a/N$ exceeds a specific threshold with other factors (e.g., $N$ and $G$) remains unchanged, the expert size increases progressively while the number of experts decreases. It induces gradual structural distortion in the MoE architecture, which in turn disrupt the advantage of MoE's combinational activation and thus degrades the performance. We have also validated that Eq. 7 can accurately predict the loss for models with larger model sizes. More discussions and detailed parameter fitting procedures are given in Appendix C.3 and F.

### 4.3 IMPACT OF THE NUMBER OF ACTIVATED EXPERTS $G$

$G$ constitutes another critical factor, defined as the number of activated experts (including activated shared and routed experts). It reflects the granularity of expert partitioning in MoE architectures. To investigate the scaling law with $G$ as the independent factor, a series of controlled experiments were conducted with other factors constant, shown in Figure 3 and Figure 11. With the increase in $G$, the validation loss exhibits a trend of first decreasing and then increasing. It is hypothesized that the impact of $G$ better conforms to a hook function relationship, expressed as follows:

$$L(G) = eG + \frac{f}{G} + \tau. \tag{8}$$

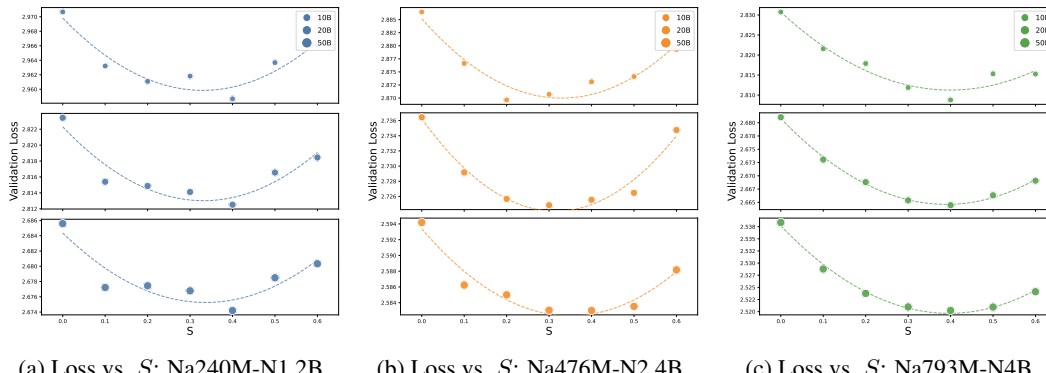

(a) Loss vs. $S$: Na240M-N1.2B.  (b) Loss vs. $S$: Na476M-N2.4B.  (c) Loss vs. $S$: Na793M-N4B.

Figure 4: Marginal effects of validation loss with respect to $S$. (a), (b) and (c) respectively illustrate the marginal relations between loss and $S$ under different $N$, $D$ settings. For small $N$ and $D$, the trend of $S$ is taking shape but is also easily affected by noise. As they increase, the scaling law of $S$ gradually becomes significant and robust. More details are in Appendix C.5.

It is noteworthy that the possible $G$'s exponent term approaches 1. Therefore, based on the fitting results and the Occam's Razor principle (Blumer et al., 1987), the exponent term of $G$ is omitted.

### 4.4 THE JOINT MOE SCALING LAW OF $N$, $D$, $N_a$ AND $G$

Based on the above conclusions in Eq. 7 and 8, we explore the variation patterns of the fitted hyperparameters $a, \alpha, b, \beta, c, h, \epsilon$ under different values of $G$. We observe that $a$, $c$ and $h$ exhibit the hook-function trend with variations in $G$, whereas other hyperparameters are largely unaffected by $G$ and display no discernible pattern in Figure 10. Hence, we can express them as $a = e_1 G + \frac{f_1}{G} + \tau_1$, $c = e_2 G + \frac{f_2}{G} + \tau_2$ and $h = e_3 G + \frac{f_3}{G} + \tau_3$. Furthermore, after re-parameterizing them, we notably found that $(e_1, f_1)$, $(e_2, f_2)$ and $(e_3, f_3)$ exhibit a proportional correlation and $\tau_3 \approx 0$. Therefore, the scaling law for $L(N, D, N_a, G)$ is presented as follows:

$$L(N, D, N_a, G) = (eG + \frac{f}{G}) * (\frac{1}{N^\alpha} + \frac{k}{N_a^\alpha} + h\frac{N_a}{N}) + \frac{a}{N^\alpha} + \frac{b}{D^\beta} + \frac{c}{N_a^\alpha} + \epsilon. \quad (9)$$

Here, $\frac{a}{N^\alpha} + \frac{b}{D^\beta} + \epsilon$ denotes the basic Chinchilla-like scaling law part from Eq. 4. Considering that $N_a$ and $N$ exhibit a similar mechanism of action on the loss to a certain extent, we also have a power-law term $\frac{c}{N_a^\alpha}$ for $N_a$. The right terms $\frac{a}{N^\alpha} + \frac{b}{D^\beta} + \frac{c}{N_a^\alpha} + \epsilon$ could be viewed to characterize the scaling law with respect to (activated/total) model size and data size. For the left term, $eG + \frac{f}{G}$ denotes the effect of G (related to MoE expert structure) on loss from Eq. 8 and such effect is scaled by the model size factors ($\frac{1}{N^\alpha} + \frac{k}{N_a^\alpha} + h\frac{N_a}{N}$, which is non-monotonic for $N_a$ introduced in Eq. 7). The detailed analysis of hyperparameters and fitting results of Eq. 9 are provided in Appendix C.4.

### 4.5 EXTENDED JOINT MOE SCALING LAW WITH SHARED EXPERT RATIO $S$

The shared experts have been verified to be essential in popular MoE architectures (Dai et al., 2024; Liu et al., 2025). We set the shared expert ratio $S$ ($S = n_s/G$, where $n_s$ represents the number of shared experts) as the sole varying factor to conduct experiments as presented in Figure 4. We find that MoE models with shared experts significantly outperform those without shared experts, verifying the necessity of shared expert isolation. As $S$ increases, the loss first decreases and then increases, while it exerts a relatively minor impact on losses around the optimal point. Based on these observations, we adopt a quadratic function to capture the marginal effect of $S$ on the loss.

$$L(S) = mS^2 + nS + \psi. \quad (10)$$

**Final Joint MoE Scaling Law.** We then incorporate $S$ into Eq. 9 to build our final joint MoE scaling law with all five factors. Given that $S$ exerts only a minor influence on loss in a wide range, we assume that $S$ has negligible impact on the form and hyperparameters. Accordingly, we perform

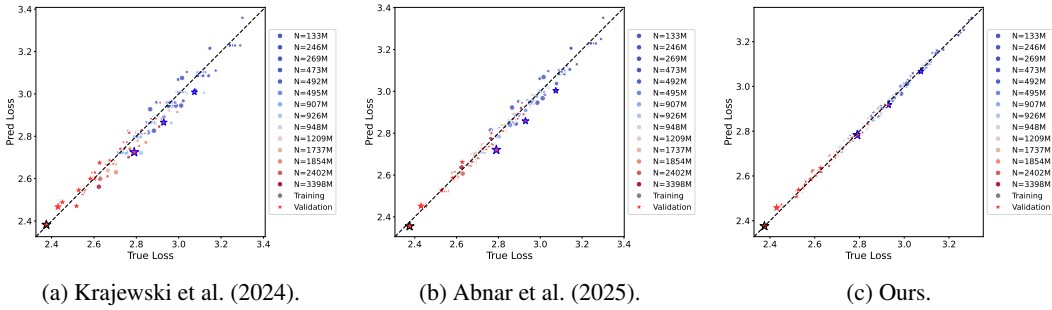

(a) Krajewski et al. (2024).  (b) Abnar et al. (2025).  (c) Ours.

Figure 5: Fitting results of existing and our joint scaling laws for MoE architectures. The average validation loss errors of (a), (b) and (c) are respectively 0.0193, 0.0170 and 0.0060. Star points are validation data with larger model sizes, larger data sizes and different MoE settings. Data point size is proportional to $D$. Detailed descriptions of compared scaling laws are provided in Appendix I.

hyperparameter fitting for Eq. 10 across diverse configurations of $N$, $D$, $N_a$ and $G$ and analyze the relationships between these fitted hyperparameters and factors. The results are presented in Figure 13 with key findings: (1) $m$ and $n$ are independent of $D$, indicating that $D$ and $S$ are mutually decoupled; (2) $m$ increases with the growth of $N$ and $N_a$, whereas $n$ decreases with the growth of $N$ and $N_a$. Notably, the extreme point of $S$ remains unchanged with variations in $N$ and $N_a$; (3) $\psi$ exhibits an obvious power-law relationship with $N$, $N_a$ and $D$; (4) As can be inferred from the definition of $S$ in Eq. 3, $S$ is already correlated with $G$, so its relationship with $G$ will not be further considered. Considering the compatibility with Eq. 9, the form of $L(N, D, N_a, G, S)$ is expressed as: $(L(G) + L(S)) * \phi(N, N_a) + \frac{a}{N^\alpha} + \frac{b}{D^\beta} + \frac{c}{N_a^\alpha} + \epsilon$. Consequently, we propose our MoE scaling law comprehensively with $N$, $D$, $N_a$, $G$, $S$ as follows:

$$L(N, D, N_a, G, S) = (eG + \frac{f}{G} + mS^2 + nS) * (\frac{1}{N^\alpha} + \frac{k}{N_a^\alpha} + h\frac{N_a}{N}) + \frac{a}{N^\alpha} + \frac{b}{D^\beta} + \frac{c}{N_a^\alpha} + \epsilon. \quad (11)$$

Note that the factors of $S$ and $G$, which jointly characterize the MoE structure of activated experts, are included in the first term, while the other terms remain consistent with those in Eq. 9. Their impact on the loss is also regulated by the total model size $N$ and the activated model size $N_a$. A detailed analysis of hyperparameters is presented in Appendix C.5.

**Fitting Results.** To determine the specific value of hyperparameters in Eq. 11, we implement all 450 experiments that encompass diverse configurations of $N$, $D$, $N_a$, $G$ and $S$. The corresponding hyperparameter values are provided in Table 2 in Appendix C.1. Next, we evaluate the fitting performance of our MoE scaling law in Figure 5, where the satisfactory fitting performance demonstrates the advantage of our MoE scaling law compared to others. Notably, for the sake of fairness, during the comparison, the baseline MoE scaling laws were first re-fitted with hyperparameters using the same data points, followed by prediction. Moreover, we extend the experiments to larger MoE models (up to 9B total model size and 100B trained tokens) with different MoE structure settings ($G$ ranges from 2 to 20 and $S$ ranges from 0 to 0.7). The consistently accurate fitting results demonstrate that our scaling law maintains robust performance when applied to larger-scale MoE models with broad ranges of parameter selections.

## 5 KEY IMPLICATIONS FOR MoE ARCHITECTURE DESIGN

In this section, we discuss the insightful findings deduced on the basis of our MoE scaling laws, which are anticipated to provide more effective guidance for the design of better MoE models.

### 5.1 IMPLICATION-1: OPTIMAL NUMBER OF ACTIVATED EXPERTS $G$

In light of Eq. 11, the optimal number of activated experts $G$ can be expressed as follows:

$$G_{opt} = \sqrt{f/e}. \quad (12)$$

Eq. 12 demonstrates that the optimal $G$ is independent of model size $N$, activated model size $N_a$ and data size $D$, thereby corresponding to a fixed optimal value $G_{opt} \approx 6.78$. Moreover,

the theoretically derived optimal $G$ exhibits strong alignment with the configurations employed by current mainstream MoE models, including DeepSeek-V3.1 (Liu et al., 2024), Kimi-K2 (Kimi et al., 2025) and Qwen3-235B-A22B (Qwen Team et al., 2025) (both $G = 8\ or\ 9$). The detailed formula derivation is provided in Appendix D.

## 5.2 IMPLICATION-2: OPTIMAL RATIO OF SHARED EXPERTS $S$

Similarly, $S$ is independent of other factors and also has an optimal value as follows:

$$S_{opt} = -n/2m. \tag{13}$$

Eq. 13 shows that the optimal $S \approx 0.31$ is also independent of other factors. As shown in Table 3, since $S$ has minor impact on loss around the optimal point as stated in Section 4.5, the appropriate $S$ values are approximately distributed in the range of $[0.13, 0.31]$ for the majority of popular MoE settings, with the loss deviation to the optimal setting's less than $0.001$. In conclusion, our findings lead to the following recommendation: the shared expert constitutes an essential component. For the optimal total activated expert number $G = 7$, we could set 1 or 2 shared experts. Notably, this aligns with the architectural configurations observed in open-source canonical MoE models, corroborating the validity of our conclusions. More details and analyses are in Appendix E.

## 5.3 IMPLICATION-3: OPTIMAL ACTIVATED PARAMETER RATIO $N_a/N$

**Theoretical Analysis.** From Eq. 11, it can be observed that there are two types of terms involving the activated model size $N_a$ as a numerator or denominator. These two terms exert opposite effects on the loss of MoE models. Intuitively, this implies that there exists an optimal $N_a$ given the configurations of other factors (e.g., the model size $N$). Given $N$, the $\left(\frac{N_a}{N}\right)_{opt_t}$ that achieves the theoretically optimal loss is formalized as follows (Comprehensive derivation is in Appendix F):

$$\left(\frac{N_a}{N}\right)_{opt_t} = \left(\frac{\alpha \cdot \left[k \cdot \left(eG + \frac{f}{G} + mS^2 + nS\right) + c\right]}{hN^\alpha \cdot \left(eG + \frac{f}{G} + mS^2 + nS\right)}\right)^{\frac{1}{\alpha+1}} = \left(\frac{\alpha \cdot [k \cdot const + c]}{hN^\alpha \cdot const}\right)^{\frac{1}{\alpha+1}}. \tag{14}$$

According to our Implications #1 and #2, $G$ and $S$ have optimal values and thus $eG + \frac{f}{G} + mS^2 + nS$ can be represented as a constant term $const$ under the optimal setting. Eq. 14 indicates that the optimal $\left(\frac{N_a}{N}\right)_{opt_t}$ decreases as the model size $N$ increases. It verifies that with the increasing total model sizes, the optimal MoE architecture will be sparser with smaller $N_a$, which is consistent with the current trend of MoE models (Kimi et al., 2025; OpenAI et al., 2025). For instance, for $N$ from 30B (Qwen3-30B-A3B (Qwen Team et al., 2025)) to 671B (Deepseek-V3.1 (Liu et al., 2024)), the theoretically optimal ratio satisfies $\left(\frac{N_a}{N}\right)_{opt_t}$ range from $40.0\%$ to $22.0\%$.

**Practical Efficiency-aware Analysis.** However, the theoretically optimal sparsity degree of MoE $\frac{N_a}{N}$ calculated in Eq. 14 cannot be directly used to guide the real-world MoE architecture design, as the efficiency of LLMs is also an essential factor. Specifically, when $N_a$ gradually increases toward its optimal value, the performance gains become increasingly marginal, while the associated costs rise steadily. Therefore, it is necessary to explore more practical efficiency-aware optimal $\left(\frac{N_a}{N}\right)_{opt_e}$ under the consideration of the balance between performance gain and efficiency cost.

Specifically, we define the loss gain threshold as $\Delta$Loss for the step size of $\Delta N_a$ set as $0.01N$. As $N_a$ is incrementally scaled for each step size, the marginal gain of loss reduction will ultimately fall below the loss gain threshold $\Delta$Loss, where we suppose the model reaches the practical efficiency-aware optimal $\left(\frac{N_a}{N}\right)_{opt_e}$. Comprehensive derivation and pseudo code are in Appendix F. Hence, for a given model size $N$, our MoE scaling law yields a practically applicable range for $N_a$, spanning the interval from the practical efficiency-aware optimal point to the theoretical optimal point, i.e., $N_a \in \left[\left(\frac{N_a}{N}\right)_{opt_e}, \left(\frac{N_a}{N}\right)_{opt_t}\right]$. To substantiate the validity of our conclusions, we conducted an analysis on the configurations of mainstream industrial MoE models, with detailed specifications in Table 4. It shows that the activated model sizes of most mainstream MoE models are within our recommended range above. Note that some recent MoE models (e.g., Kimi-K2 (Kimi et al., 2025) and gpt-oss-120b (OpenAI et al., 2025)) employ a more aggressive sparser architecture with $\frac{N_a}{N} \leq 4\%$, primarily aiming to reduce the training/inference costs in practice with larger total model sizes.

# 6 RELATED WORK

## 6.1 MoE ARCHITECTURE

In the field of language models, the MoE model enables experts to learn different knowledge and combine their outputs (Abdin et al., 2024; Team et al., 2025; Zeng et al., 2025; Lieber et al., 2024). Shazeer et al. (2017) expanded upon this with the Sparsely-Gated Mixture-of-Experts (SMoE) layer and Top-$K$ routing, which selects a fixed number of experts for each token. This was further developed by Gshard (Lepikhin et al., 2020) and SwitchTransformer (Fedus et al., 2022) by integrating MoE into Transformer feedforward layers with Top-1 and Top-2 routing. More recently, Dai et al. (2024) proposed modifying the MoE layer by subdividing experts into smaller experts and adding shared experts into the architecture. These advancements continue to enhance the efficiency and flexibility of MoE. At the same time, there is a trend toward scaling MoE to larger model sizes and to a greater number of experts, as shown in recent works such as K2 (Kimi et al., 2025), DeepSeek-V3 (Liu et al., 2024) and Mixture of a Million Experts (He, 2024). In parallel, several studies have investigated alternative routing strategies and expert designs, including heterogeneous experts in HMoE (Wang et al., 2024a), autonomous expert activation in AoE (Lv et al., 2025) and probabilistic Top-$P$ routing (Zhou et al., 2022). In this work, when analyzing scaling laws, we adopt the classical Top-$K$ routing strategy as our main setting.

## 6.2 SCALING LAWS OF LLMS

Scaling laws for LLMs describe how model performance depends on factors such as model size and training data. In dense Transformers, Kaplan et al. (2020) first studied scaling laws and showed that the final model perplexity follows a power-law relationship with both model size and data size. Building on this, Hoffmann et al. (2022) extended the analysis by incorporating variable cosine cycle lengths and proposed a revised scaling formulation. Scaling behavior has also been examined in alternative architectures and training regimes, particularly in MoE models. Clark et al. (2022) investigated MoE scaling laws under a fixed dataset, focusing on the impact of model size and the number of experts. Krajewski et al. (2024) studied how scaling changes with different levels of expert granularity in MoE architectures. Abnar et al. (2025) analyzed scaling laws with respect to total model size $N$, dataset size $D$ and the fraction of inactive experts, while Ludziejewski et al. (2025) examined the joint effects of multiple factors, including the activated model size $N_a$, dataset size $D$. The comparison between these scaling laws and ours is in Table 5.

Beyond architectures, Step Law (Li et al., 2025a) and relevant studies (Shuai et al., 2024; Zhang et al., 2024; McCandlish et al., 2018) provide principles for learning rate, weight decay and batch size, while Farseer (Li et al., 2025b) refines loss scaling for accurate extrapolation. Other works include parallel scaling (Chen et al., 2025) for improving compute efficiency and SynthLLM (Qin et al., 2025) for analyzing the scalability of synthetic data. More recently, Sun et al. (2025) proposed a joint scaling law for floating point quantization training, highlighting the influence of exponent and mantissa bits, the effect of critical data size and the optimal precision range for efficient LLM training. Overall, these works extend scaling laws across optimizers, architectures and data, providing guidance for LLM design and training. Nevertheless, the community is urgent to build a more comprehensive MoE scaling law.

# 7 CONCLUSION AND FUTURE WORK

In this work, we propose a more accurate joint MoE scaling law that considers more comprehensive factors, including total model size $N$, data size $D$, activated model size $N_a$, the number of activated experts $G$ and the ratio of shared experts in activated experts $S$. Based on the joint MoE scaling law, we further derive the optimal value expressions for essential MoE-specific factors of $G$, $S$ and $N_a/N$ and identify several insightful implications, which can facilitate future MoE model design and training. In the future, we will further validate our scaling law under larger scales and novel MoE architectures. Currently, our investigations have primarily centered on factors pertaining to the MoE blocks. It would be worthwhile to extend this scope to encompass both the factors and structural configurations associated with other LLM blocks, such as the attention layers.

## REPRODUCIBILITY STATEMENT

To ensure the reproducibility of our results, we have taken the following measures: (1) **Datasets**. In Section 3, we specify the dataset (Dolma V1.7 dataset) used in the experiments and provide its source attribution. (2) **Code**. All experiments in our scaling law research were trained using open-source frameworks (Megatron and Torchtitan), ensuring high reproducibility. Details are provided in the supplementary materials. (3) **Experimental Details**. The basic experimental settings are described in Section 3, with specific hyperparameter configurations provided in Appendix B and detailed experimental specifications in Appendix L. (4) **Proof Details**. The derivation process of our joint MoE scaling law is elaborated in detail in Section 4 and Appendix C. Meanwhile, the derivation of the key implications involved is thoroughly explained in Section 5, as well as in Appendices D, E, F and G.

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

## A    THE USE OF LARGE LANGUAGE MODELS (LLMS)

In this paper, we leveraged LLMs to support and refine the writing. Specifically, LLMs were used for grammar and spelling correction, as well as polishing linguistic expressions to enhance clarity and readability. All other core components of the work, including the development of ideas, design and execution of experiments and derivation of formulas, were completed manually by ourselves.

## B    HYPERPARAMETER DETAILS

We report serveral typical hyper-parameters used for training our MoE models in Table 1. The models vary in layers, hidden size and expert size across different scales, while the optimizer and learning rate settings are consistent. All models are trained with AdamW and cosine learning rate decay, using a sequence length of 2048 and a batch size of 2M tokens. The detailed hyper-parameters of our MoE models are given as follows. In all of our experiments, the parameters considered do not include those from the embedding layer. For all experimental settings, refer to Appendix L.

Table 1: Typical model hyper-parameters for different sizes.

| Total model size | 247M | 496M | 907M | 2.40B | 3.96B |
|---|---|---|---|---|---|
| Activated model size | 48M | 99M | 181M | 476M | 793M |
| # Layers | 12 | 12 | 12 | 20 | 24 |
| # Routed experts | 32 | 32 | 32 | 32 | 32 |
| # Activated routed experts | 4 | 4 | 4 | 4 | 4 |
| # Shared experts | 1 | 1 | 1 | 1 | 1 |
| # Attention heads | 8 | 12 | 16 | 20 | 24 |
| Hidden size | 512 | 768 | 1024 | 1280 | 1536 |
| Expert size | 384 | 512 | 704 | 896 | 1024 |
| Attention head size | 64 | 64 | 64 | 64 | 64 |
| Optimizer | AdamW | | | | |
| Adam $(\beta_1, \beta_2)$ | (0.9, 0.95) | | | | |
| Adam $\epsilon$ | $1 \times 10^{-8}$ | | | | |
| Weight decay | 0.1 | | | | |
| Clip grad norm | 1.0 | | | | |
| Max lr | $3.0 \times 10^{-4}$ | | | | |
| Min lr | 0 | | | | |
| Lr decay | Cosine | | | | |
| Decay rate | 10% | | | | |
| Sequence length | 2048 | | | | |
| Batch size (# tokens) | 2M | | | | |
| Warmup steps | 500 | | | | |
| Normalization | RMSNorm | | | | |
| Vocabulary size | 128256 | | | | |
| Positional encoding | ROPE | | | | |

## C    FITTING DETAILS OF OUR MOE SCALING LAWS

Our MoE scaling law precisely characterizes the effects of three critical dimensions—parameters, data sizes and model architectures—on scaling patterns. Specifically, the term $L(G, S) \cdot \phi(N_a, N)$ quantifies the architectural impact on loss, while explicitly revealing its regulation by parameter scale. The formulation $\rho(N, D, N_a) = \frac{a}{N^\alpha} + \frac{b}{D^\beta} + \frac{c}{N_a^\alpha} + \epsilon$ delineates how parameters and data size influence the MoE loss. By integrating all core factors that govern MoE architectural performance into a joint scaling law, our proposed MoE scaling law achieves an elegant integration. This theoretical construct carries substantial significance for informing the design of MoE model architectures. Further details regarding the fitting process of our MoE scaling laws are elaborated below.

## C.1 NUMBERICAL FITS OF OUR JOINT MOE SCALING LAW

Our joint MoE scaling law is formalized as:

$$L(N, D, N_a, G, S) = (eG + \frac{f}{G} + mS^2 + nS) * (\frac{1}{N^\alpha} + \frac{k}{N_a^\alpha} + h\frac{N_a}{N}) + \frac{a}{N^\alpha} + \frac{b}{D^\beta} + \frac{c}{N_a^\alpha} + \epsilon, \quad (15)$$

where the detailed fitted constants and values are presented below, based on 450 experiments across different model settings. Of these, 268 were used for fitting, 91 for validation, and 90 small-size experiments to observe the marginal effect of $G$.

Table 2: Fitted constants and their values in Eq. 15.

| Constant | Value |
|---|---|
| $e$ | 0.1577 |
| $f$ | 7.2446 |
| $m$ | 5.1395 |
| $n$ | -3.2363 |
| $k$ | 0.0013 |
| $h$ | 0.0450 |
| $a$ | 38.0510 |
| $\alpha$ | 0.2383 |
| $b$ | 27129.0488 |
| $\beta$ | 0.4694 |
| $c$ | 31.0958 |
| $\epsilon$ | 1.8182 |

## C.2 FITTING RESULTS OF THE SCALING LAW FOR $L(N, D)$

Figure 6 presents the fitting performance of the $L(N, D)$ Eq. 4 in fitting the loss of MoE architectures, where only total model size $N$ and data size $D$ are considered. It can be observed that $L(N, D)$ only provides a coarse-grained fit to our experimental data points, with suboptimal specific fitting performance. This indicates that additional factors within the MoE architecture need to be taken into account.

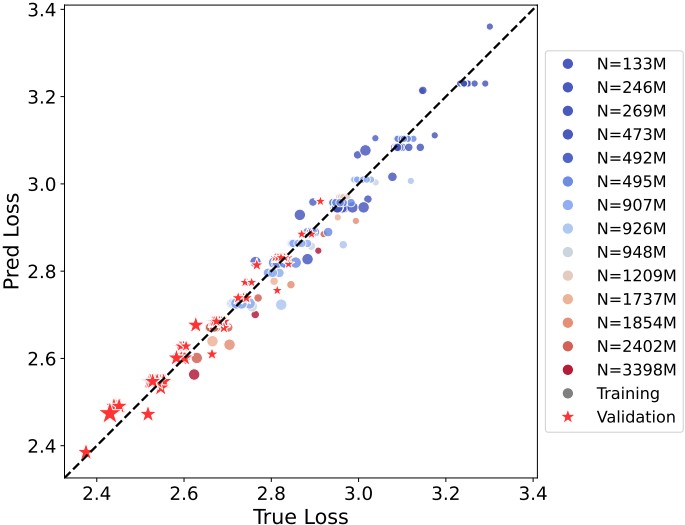

Figure 6: Fitting results of the scaling law of $L(N, D)$. Average validation loss error: 0.0179.

C.3 FITTING RESULTS OF THE SCALING LAW FOR $L(N, D, N_a)$

Figure 7 demonstrates the fitting performance of the scaling law $L(N, D, N_a)$ Eq. 7 on the loss of MoE models, where $N_a$ is incorporated into the law. This result indicates that $N_a$ constitutes a critical factor influencing the MoE scaling law and the inclusion of $N_a$ in the joint scaling law yields a substantial improvement in fitting performance compared to the original scaling law $L(N, D)$.

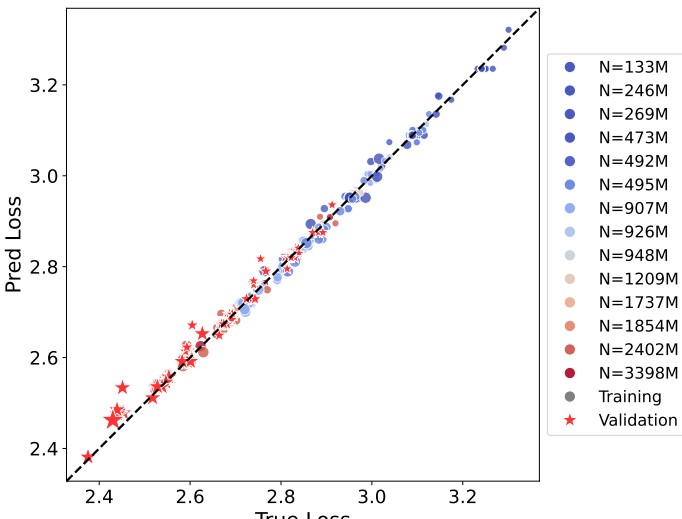

Figure 7: Fitting results of the scaling law of $L(N, D, N_a)$. Average validation loss error: 0.0124.

In the following, we elaborate on the process of incorporating the factor $N_a$ into the scaling law $L(N, D, N_a)$ Eq. 7. Specifically, we first designed and conducted a series of controlled experiments on $N_a$ following Eq. 35. Subsequently, hyperparameter fitting was performed across diverse configurations of $D$ and $N$, with the associated results presented in Figure 8.

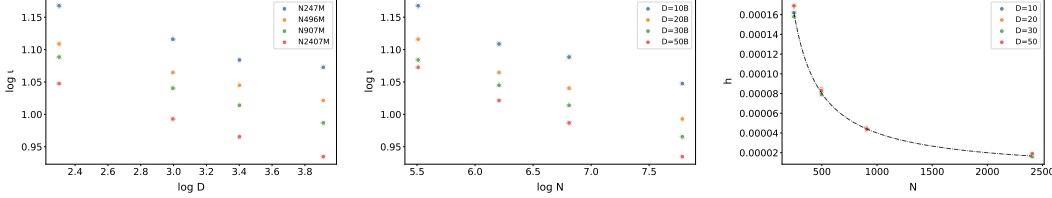

Figure 8: The correlations between $\iota$, $h$ in Eq. 5 and $N$, $D$. $\iota$ and $h$ can be viewed as functions of $N$ or $D$.

It can be observed that $\iota$ exhibits a linear relationship with $N$ and $D$ after logarithmic transformation, indicating a power-law relationship between $\iota$ and $N$ as well as $D$. The parameter $h$ shows an inverse proportional relationship with $N$ across different data volumes $D$. Other factors $c$ and $\gamma$ fluctuate with changes in $N$ and $D$ without displaying obvious correlations and thus are considered independent of $N$ and $D$. It is noteworthy that the hyperparameter fitting results indicate the fitted value of the exponent term for $N_a$ in the term $h\frac{N_a}{N}$ approaches 1. Therefore, it is reasonable to conclude that $N_a$ in the term $h\frac{N_a}{N}$ does not have an exponent.

C.4 FITTING RESULTS OF THE SCALING LAW FOR $L(N, D, N_a, G)$

Figure 9 illustrates the fitting performance of the scaling law $L(N, D, N_a, G)$ Eq. 9 on the loss of MoE architectures, where the granularity $G$ is incorporated. As stated in Section 4.3, $G$ characterizes

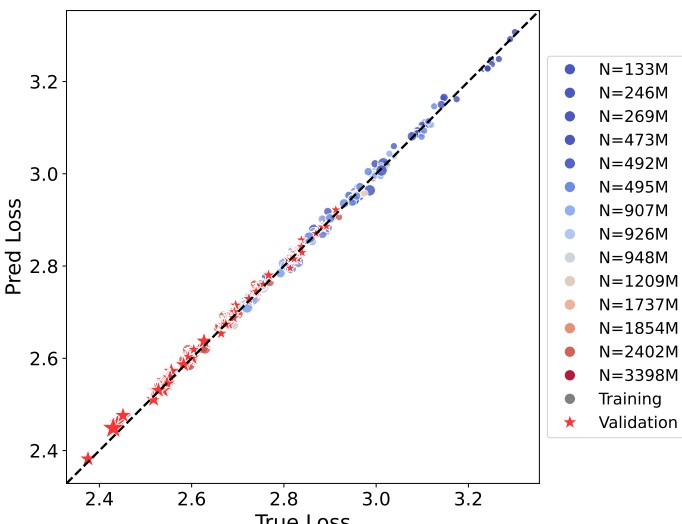

Figure 9: Fitting results of the scaling law of $L(N, D, N_a, G)$. Average validation loss error: 0.0083.

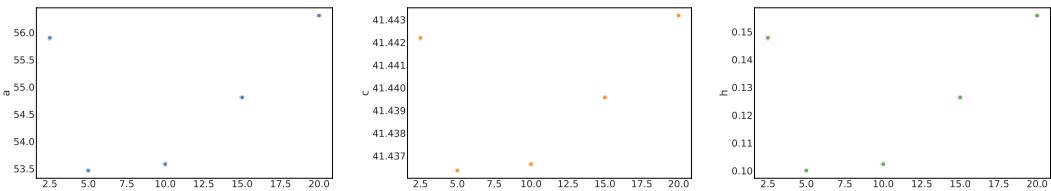

Figure 10: The correlations between $a, c, h$ in Eq. 7 and G. $a, c, h$ can be viewed as functions of G.

the structural properties of MoE architectures and reflects the impact of MoE architecture on performance. The results demonstrate that the scaling law which considers the structural factor $G$ yields superior fitting performance. By analogy, to incorporate $G$ into the scaling law $L(N, D, N_a, G)$, we first designed and conducted a series of controlled experiments where $G$ was treated as the sole factor of variation, with all other factors held constant. Specifically, as $G$ increases, the counts of routed experts and shared experts expand proportionally, whereas the corresponding expert dimensions shrink proportionally. Considering the marginal effect between $G$ and loss, as well as the coupling relationships among $G$, $N$ and $N_a$, we hypothesize that $a$, $b$, $c$ and $h$—which appear in the numerator—are correlated with $G$. Experimental results demonstrate that the data size $D$ is independent of $G$. The variation curves of the fitted hyperparameters $a, c, h$ under different values of $G$ are presented in Figure 10. Furthermore, we also provide the scaling law of validation loss with respect to $G$ under other settings, which serve to observe the marginal effect of $G$.

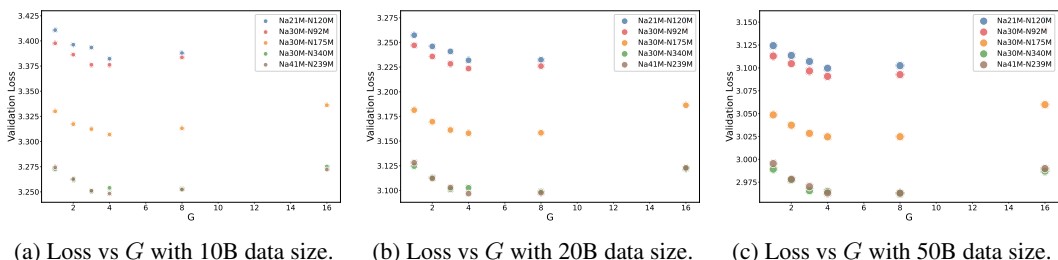

(a) Loss vs $G$ with 10B data size. (b) Loss vs $G$ with 20B data size. (c) Loss vs $G$ with 50B data size.

Figure 11: Marginal effect of validation loss and $G$ with $S = 0$. Data point sizes are proportional to $D$.

### C.5   Fitting Results of the Scaling Law for $L(N, D, N_a, G, S)$

Similarly, the ratio of shared experts to activated experts ($S$) constitutes another critical structural characteristic of MoE architectures. The fitting performance of our final joint scaling law $L(N, D, N_a, G, S)$ is illustrated in Figure 12.

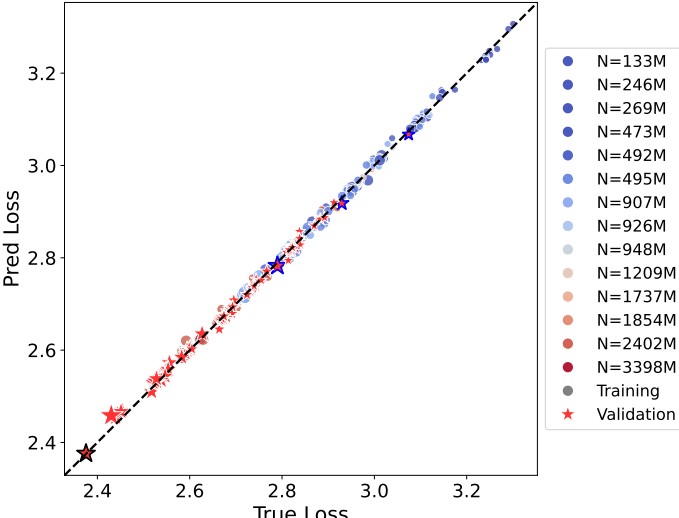

Figure 12: Fitting results of our final scaling law of $L(N, D, N_a, G, S)$. Average validation loss error: 0.0059.

Similarly, the analysis of hyperparameters related to $S$ is presented in Figure 13. It can be observed: (1) $m$ and $n$ are independent of $D$, indicating that $D$ and $S$ are mutually decoupled; (2) $m$ increases with the growth of $N$ and $N_a$, whereas $n$ decreases with the growth of $N$ and $N_a$. Notably, the extreme point of $S$ remains unchanged with variations in $N$ and $N_a$; (3) $\psi$ exhibits an obvious power-law relationship with $N$, $N_a$ and $D$.

Furthermore, as stated in Section 4.5, the scaling law of $S$ becomes increasingly prominent with the growth of model size. As shown in Figure 14.

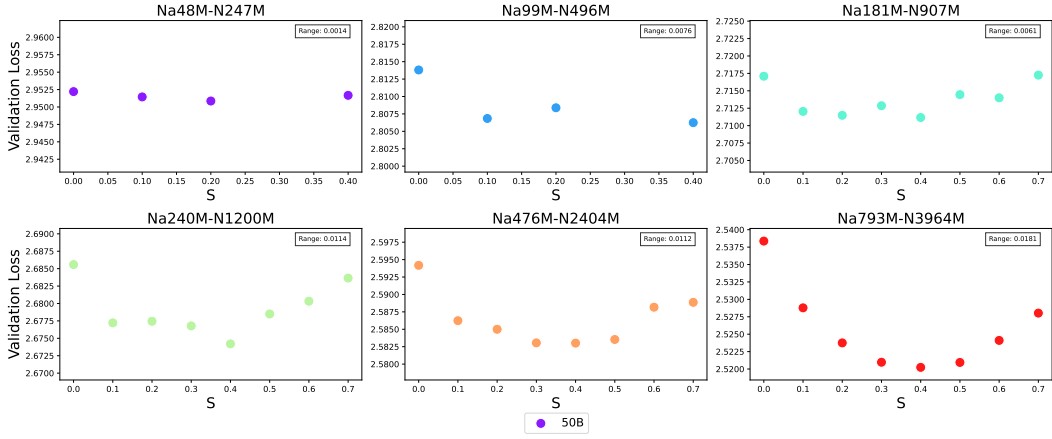

Figure 14: Illustration of the scaling law of $S$ becoming increasingly prominent with growing model size with a data size of 50B.

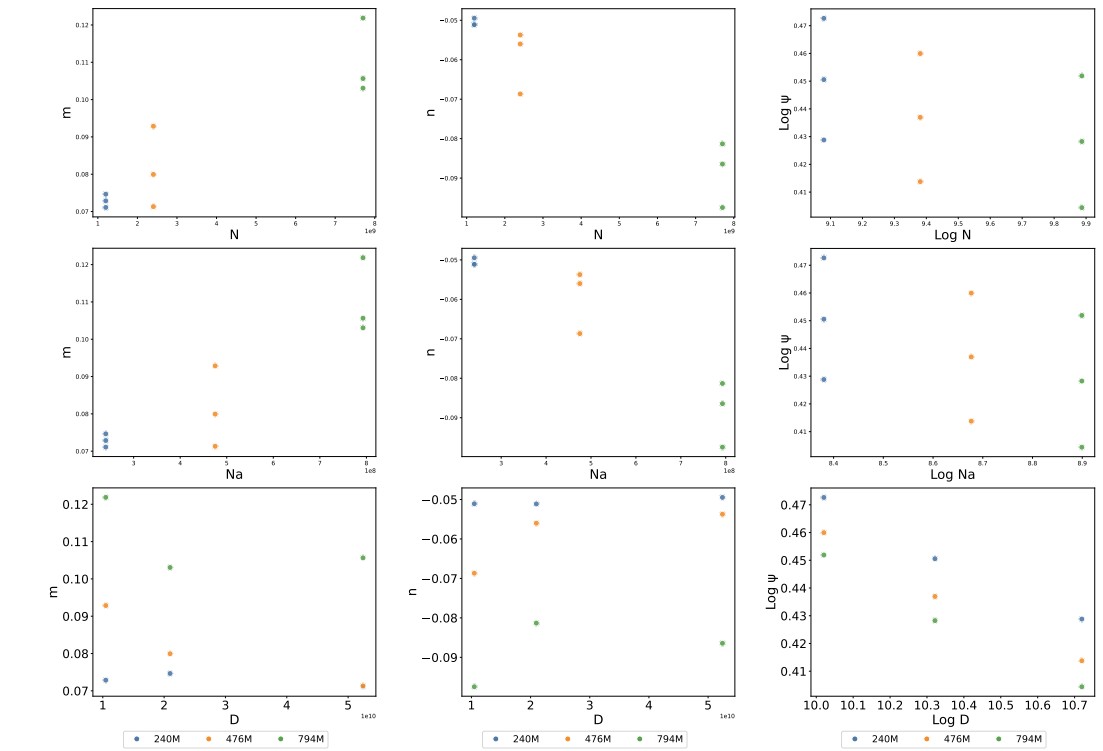

Figure 13: The correlations between $m$, $n$ and $\psi$ in Eq. 10 and $N$, $N_a$, $D$. $m$, $n$ and $\psi$ can be viewed as functions of $N$, $N_a$ or $D$.

## D    DERIVATION OF THEORETICAL OPTIMAL $G$

First, we decompose $L(N, D, N_a, G, S)$ Eq. 15 into three components to isolate terms involving $G$. This decomposition leverages the fact that most variables ($N, D, N_a, S$) are independent of $G$:

$$L_{N,D,N_a,S}(G) = \underbrace{\left(eG + \frac{f}{G} + mS^2 + nS\right)}_{A(G)} \cdot \underbrace{\left(\frac{1}{N^\alpha} + \frac{k}{N_a^\alpha} + h\frac{N_a}{N}\right)}_{B}$$
$$+ \underbrace{\left(\frac{a}{N^\alpha} + \frac{c}{N_a^\alpha} + \frac{b}{D^\beta} + s\right)}_{C} = A(G) \cdot B + C \tag{16}$$

Compute $\frac{\partial L}{\partial G}$ and set to 0:

$$\frac{\partial L}{\partial G} = \left(e - \frac{f}{G^2}\right) \cdot B = 0 \tag{17}$$

Since $B \neq 0$, we get $e - \frac{f}{G^2} = 0$. Therefore:

$$G^2 = \frac{f}{e} \implies G = \sqrt{\frac{f}{e}} \quad (G > 0) \tag{18}$$

The second derivative checks to confirm a minimum.

$$\frac{\partial^2 L}{\partial G^2} = \frac{2f}{G^3} \cdot B > 0 \tag{19}$$

The extreme point (minimum) is:

$$G_{opt} = \sqrt{\frac{f}{e}} \tag{20}$$

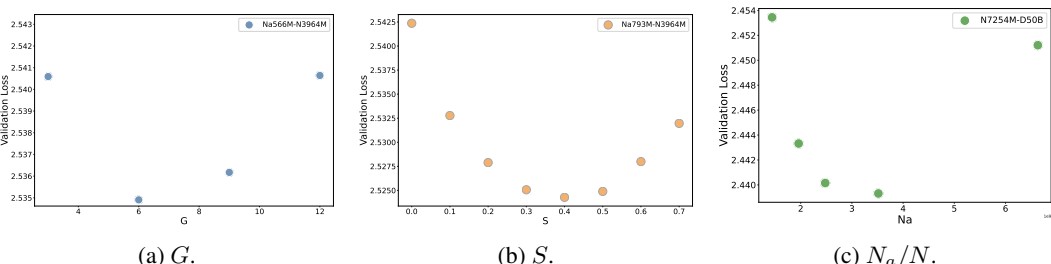

(a) $G$.  (b) $S$.  (c) $N_a/N$.

Figure 15: Verification of the marginal relationship for $G$, $S$ and $N_a/N$ under larger model sizes.

Substituting the optimized hyperparameters into the above Eq. 20 yields $G_{\text{opt}} \approx 6.78$.

To substantiate the validity of our conclusions, we conducted an analysis on the configurations of mainstream industrial MoE models, with detailed specifications in Table 3.

Table 3: Analyses of mainstream MoE models. Based on our scaling law, the theoretical optimal $G$ and $S$ are set as: $G \approx 6.78$, $S \approx 0.31$. The practical ranges provide relative recommended settings of $G$ and $S$ in practice with less effectiveness loss ($\leq 0.001$).

| Model | G ($n_s$+TopK) | G (Actual) | G Practical Range (Thr = 0.001) | S (Actual) | S Practical Range (Thr = 0.001) |
|---|---|---|---|---|---|
| gpt-oss-20b | $0+4$ | 4 | $[5.09, 9.04]$ | 0 | $[0.183, 0.446]$ |
| Qwen3-30B-A3B | $0+8$ | 8 | $[4.80, 9.58]$ | 0 | $[0.156, 0.473]$ |
| Hunyuan-A13B | $1+8$ | 9 | $[4.99, 9.21]$ | 1/9 | $[0.175, 0.455]$ |
| GLM-4.5-Air | $1+8$ | 9 | $[4.77, 9.64]$ | 1/9 | $[0.154, 0.476]$ |
| gpt-oss-120b | $0+4$ | 4 | $[4.27, 10.77]$ | 0 | $[0.102, 0.528]$ |
| Qwen3-235B-A22B | $0+8$ | 8 | $[4.61, 9.98]$ | 0 | $[0.138, 0.492]$ |
| GLM4.5 | $1+8$ | 9 | $[4.56, 10.09]$ | 1/9 | $[0.133, 0.497]$ |
| Deepseek-V3.1 | $1+8$ | 9 | $[4.20, 10.93]$ | 1/9 | $[0.095, 0.535]$ |
| Kimi-K2 | $1+8$ | 9 | $[3.85, 11.95]$ | 1/9 | $[0.053, 0.577]$ |

Theoretically, the optimal values of $G$ and $S$ are determined to be 6.78 and 0.31, respectively. In practical deployment scenarios, however, a trade-off range for $G$ and $S$ is typically adopted, mainly owing to inherent efficiency constraints of MoE models. Building upon our proposed joint MoE scaling law, we further derive efficiency-aware ranges for $G$ and $S$ tailored to mainstream MoE models, with the loss threshold constrained to 0.001.

As presented in Table 3, the practical ranges of $G$ exhibit a predominant distribution within the interval [4, 11] across diverse model configurations. Moreover, these practical ranges of $G$ are demonstrated to be dependent on parameters $N$ and $N_a$. Notably, the theoretically optimal value of $G$ (i.e., 7) and our recommended practical range show strong consistency with the parameter settings of mainstream MoE models across varying model scales, which implicitly corroborates the validity of our inferences regarding $G$. The verification of the marginal effect for $G$ under larger model sizes is illustrated in Figure 15(a).

# E  DERIVATION OF THEORETICAL OPTIMAL $S$

Analogous to the derivation of the optimal value of $G$, the isolation of $S$ from other factors followed by the computation of its first-order derivative $\frac{\partial L}{\partial S}$—with the derivative set to 0:

$$\frac{\partial L}{\partial S} = (2mS + n) \cdot B = 0 \tag{21}$$

Since $B \neq 0$, we get $2mS + n = 0$. Therefore:

$$S_{opt} = -\frac{n}{2m} \tag{22}$$

Likewise, the second-derivative test confirms that $S_{opt}$ corresponds to a minimum value. Substituting the optimized hyperparameters into the above Eq. 22 yields $S_{\text{opt}} \approx 0.31$.

Similarly, as shown in Table 3, we have also deduced the theoretical optimal value of $S$ and its corresponding efficiency-aware practical range. It is of particular note that both our experimental findings and deductive inferences collectively highlight the indispensable role of shared experts in the design of MoE architectures.

Existing researches on the parameter $S$ in mainstream models remain relatively insufficient. Most studies merely focus on the issue of whether to incorporate shared experts and no consistent consensus has been established in the field thus far. Our empirical findings demonstrate that the inclusion of $S$ outperforms its exclusion in terms of model performance. Notably, within a specific range of $S$ values, the model can consistently achieve satisfactory results with negligible performance fluctuations.

The verification of the marginal law for $S$ under larger model sizes is illustrated in Figure 15(b).

## F   DERIVATION OF THEORETICAL AND PRACTICAL OPTIMAL $N_a/N$

**Theoretically Analysis.** From Eq. 11, we can observe that there are two types of terms involving the activated model size $N_a$ as a numerator or denominator. These two terms exert opposite effects on the loss of MoE models. Intuitively, this implies that there exists an optimal $N_a$ given the configurations of other factors (e.g., the model size $N$). It has been experimentally validated on both the Torchtitan Liang et al. (2024) and Megatron Shoeybi et al. (2019) pre-training frameworks. To find the optimal point of $(\frac{N_a}{N})_{opt_t}$ in $L(N, D, N_a, G, S)$, firstly, let $r = \frac{N_a}{N}$, then decompose $L$ as:

$$L_{N,D,G,S}(r) = \underbrace{\left(eG + \frac{f}{G} + mS^2 + nS\right)}_{A} \cdot \underbrace{\left(\frac{1}{N^\alpha} + \frac{k}{(rN)^\alpha} + hr\right)}_{B(r)}$$
$$+ \underbrace{\left(\frac{a}{N^\alpha} + \frac{c}{(rN)^\alpha} + \frac{b}{D^\beta} + s\right)}_{C(r)} \tag{23}$$

Compute first derivative $\frac{\partial L}{\partial r}$ and set to 0:

$$\frac{\partial L}{\partial r} = A \cdot \frac{\partial B}{\partial r} + \frac{\partial C}{\partial r} = 0 \tag{24}$$

Calculate partial derivatives:

$$\frac{\partial B}{\partial r} = -\frac{\alpha k}{N^\alpha r^{\alpha+1}} + h, \quad \frac{\partial C}{\partial r} = -\frac{\alpha c}{N^\alpha r^{\alpha+1}} \tag{25}$$

Substitute and simplify:

$$A\left(-\frac{\alpha k}{N^\alpha r^{\alpha+1}} + h\right) - \frac{\alpha c}{N^\alpha r^{\alpha+1}} = 0 \tag{26}$$

Rearrange terms to isolate $r$:

$$Ah = \frac{\alpha}{N^\alpha r^{\alpha+1}}(Ak + c) \implies r^{\alpha+1} = \frac{\alpha(Ak + c)}{AhN^\alpha} \tag{27}$$

Thus:

$$r = \left(\frac{\alpha(Ak + c)}{AhN^\alpha}\right)^{\frac{1}{\alpha+1}} \tag{28}$$

For $\alpha > 0$, $\frac{\partial^2 L}{\partial r^2} = \frac{\alpha(\alpha+1)(Ak+c)}{N^\alpha r^{\alpha+2}} > 0$, confirming a minimum.

Therefore, the optimal point of $(\frac{N_a}{N})_{opt_t}$ is:

$$\left(\frac{N_a}{N}\right)_{opt_t} = \left(\frac{\alpha \cdot \left[k \cdot \left(eG + \frac{f}{G} + mS^2 + nS\right) + c\right]}{hN^\alpha \cdot \left(eG + \frac{f}{G} + mS^2 + nS\right)}\right)^{\frac{1}{\alpha+1}} \tag{29}$$

From the expression of $\left(\frac{N_a}{N}\right)_{opt_t}$, it can be deduced that the theoretical optimal value is collectively determined by factors $G$, $S$ and $N$. Specifically, $G$ (refer to Appendix D) and $S$ (refer to Appendix E), as validated by the foregoing analysis, generally exhibit respective independent optimal values. Therefore, the optimal $\left(\frac{N_a}{N}\right)_{opt_t}$ decreases as the model size $N$ increases. It verifies that with the increasing total model sizes, the optimal MoE architecture will be sparser, which is consistent with the current trend of current industry-level MoE models (Kimi et al., 2025; OpenAI et al., 2025).

**Practical Efficiency-aware Analysis.** However, the theoretically optimal sparsity degree of MoE $\frac{N_a}{N}$ calculated in Eq. 29 cannot be directly used to guide the real-world MoE architecture design, for the efficiency of LLMs is also an essential factor. Specifically, when $N_a$ gradually increases toward its optimal value, the performance gains become increasingly marginal, while the associated costs rise steadily. Therefore, it is necessary for us to explore the optimal $\frac{N_a}{N}$ under the consideration of the balance between performance gain and efficiency cost.

We define the loss gain threshold as $\Delta$Loss for the step size of $\Delta N_a$ set as $0.01N$. As $N_a$ is incrementally scaled according to the specified step size, the marginal gain of loss reduction will ultimately fall below the defined threshold $\Delta$Loss, where we suppose the model reaches the practical efficiency-aware optimal $\frac{N_a}{N}$. The detailed derivation proceeds as follows:

---

**Algorithm 1:** Find Efficiency-aware Optimal $N_a$

**Function** FindEfficiencyAwareNa($N$, $D$, $G$, $S$, *threshold*) **:**
    step $= 0.01 \times N$;
    $N_{a\_prev} \leftarrow$ step;
    loss_prev $\leftarrow \mathcal{L}(N, D, N_{a\_prev}, G, S)$;
    iteration $= 1$;
    max_iterations $= n$;
    **while** iteration $\leq$ max_iterations **do**
        $N_{a\_current} \leftarrow N_{a\_prev} +$ step;
        loss_current $\leftarrow \mathcal{L}(N, D, N_{a\_current}, G, S)$;
        loss_reduction $\leftarrow$ loss_prev $-$ loss_current;
        **if** loss_reduction $<$ *threshold* **then**
            **return** $N_{a\_current}$;
        $N_{a\_prev} \leftarrow N_{a\_current}$;
        loss_prev $\leftarrow$ loss_current;
        iteration $\leftarrow$ iteration $+ 1$;
    **return** None;

---

Hence, for a given model size $N$, our MoE scaling law yields a practically applicable range for $N_a$, i.e., spanning the interval from the practical efficiency-aware optimal point to the theoretically optimal point, i.e., $N_a \in [\left(\frac{N_a}{N}\right)_{opt_e}, \left(\frac{N_a}{N}\right)_{opt_t}]$.

To substantiate the validity of our conclusions, we conducted an analysis on the configurations of mainstream industrial MoE models, with detailed specifications in Table 4. It indicates that the activated model sizes $N_a$ of most mainstream MoE models are consistent with our recommended ranges above. Practical MoE designs could jointly consider both effectiveness and efficiency with the help of our proposed MoE scaling laws. The verification of the marginal law for $N_a/N$ under larger model sizes is illustrated in Figure 15(c).

**Discussion on the Optimal $G$ and $N_a$.** We attempt to explain possible misunderstandings that the phenomenon reflected by our MoE scaling law (i.e., $G$ and $N_a$ have optimal value) seems to "conflict with" our intuitive cognition on expanding MoE experts or activated parameters (i.e., larger $G$ and $N_a$ are likely to achieve better results). Precisely, it intuitively seems that larger values of $G$ and $N_a$ would be preferable. This notion, however, does not contradict our proposed MoE scaling law: typically, when adjusting $N_a$ while keeping the total model size $N$ fixed (which is the most natural operation of "increasing $N_a$"), such adjustment is achieved by increasing $G$, leading to the concurrent growth of both parameters $N_a$ and $G$. In this situation, our formula correctly reflects that the loss generally decreases under these circumstances. Nevertheless, when focusing on $G$ with other factors held constant, increasing $G$ results in a greater number of routed experts but progressively

Table 4: Theoretical and practical efficiency-aware optimal $N_a/N$ analysis for mainstream MoE models

| Model | Na-N (Actual) | Na/N Theoretical Opt | Na/N Practical Opt ($\Delta$Loss = 0.001) | Na/N Practical Opt ($\Delta$Loss = 0.005) |
|---|---|---|---|---|
| gpt-oss-20b | 3.6B-21B | 42.89% (9.0B) | 22.00% (4.6B) | 9.00% (1.9B) |
| Qwen3-30B-A3B | 3B-30B | 40.04% (12.0B) | 21.00% (6.3B) | 9.00% (2.7B) |
| Hunyuan-A13B | 13B-80B | 33.16% (26.5B) | 18.00% (14.4B) | 7.00% (5.6B) |
| GLM-4.5-Air | 12B-106B | 31.41% (33.3B) | 17.00% (18.0B) | 7.00% (7.4B) |
| gpt-oss-120b | 5.1B-117B | 30.82% (36.1B) | 16.00% (18.7B) | 7.00% (8.2B) |
| Qwen3-235B-A22B | 22B-235B | 26.95% (63.33B) | 14.00% (32.9B) | 6.00% (14.1B) |
| GLM-4.5 | 32B-355B | 24.89% (88.4B) | 13.00% (46.2B) | 6.00% (21.3B) |
| Deepseek-V3.1 | 37B-671B | 22.02% (147.8B) | 12.00% (80.5B) | 5.00% (33.6B) |
| Kimi-K2 | 32B-1T | 20.40% (204.0B) | 11.00% (110.0B) | 5.00% (50.0B) |

smaller expert dimensions due to partitioning. While appropriate fine-grained partitioning can enhance performance, exceeding a specific threshold will conversely impair model performance. Similarly, when focusing on $N_a$ with other factors fixed, increasing $N_a$ leads to larger expert dimensions but a reduced number of all/routed experts, thereby diminishing the sparsity advantage of the MoE model. This induces gradual structural distortion in the MoE architecture, which in turn disrupts the advantage of MoE's combinational activation mechanism and thus degrades performance.

## G  COMPUTE-OPTIMALITY WITH FIXED CONFIGURATIONS

We control the total computation cost $C = D \cdot N_a$ and analyze the relationship between the optimal loss and $C$. According to our Implications #1 and #2, $G$ and $S$ have optimal values. Thus, the term $eG + \frac{f}{G} + mS^2 + nS$ can be expressed as a constant term $const$ under the optimal configuration. When $N$ is fixed, substituting $D = \frac{C}{N_a}$ into our joint MoE scaling law (Eq. 11) yields the following result:

$$L(N_a, C) = C_0 + (const \cdot k + c) \cdot \frac{1}{N_a^\alpha} + \frac{const \cdot h}{N} \cdot N_a + \frac{bN_a^\beta}{C^\beta}$$

$$\text{where } const = eG + \frac{f}{G} + mS^2 + nS \text{ and } C_0 = \frac{const + a}{N^\alpha} + s. \tag{30}$$

To find the optimal $N_a$ (denoted $N_a^*$) that minimizes $L(N_a)$, compute the first derivative of $L(N_a)$ with respect to $N_a$ and set $\frac{dL}{dN_a} = 0$:

$$\frac{b\beta N_a^{*\beta-1}}{C^\beta} = \alpha \left(const \cdot k + c\right) N_a^{*-\alpha-1} - \frac{const \cdot h}{N} \tag{31}$$

Substitute $N_a^*$ into Eq. 30 to get the optimal loss $L^*$:

$$L^*(C) = C_0 + \frac{(const \cdot k + c)(\alpha + \beta)}{\beta} N_a^{*-\alpha} + \frac{const \cdot h(\beta - 1)}{N\beta} N_a^*,$$

$$\text{subject to } \frac{b\beta N_a^{*\beta-1}}{C^\beta} = \alpha \left(const \cdot k + c\right) N_a^{*-\alpha-1} - \frac{const \cdot h}{N}, \tag{32}$$

$$\text{where } const = eG + \frac{f}{G} + mS^2 + nS \text{ and } C_0 = \frac{const + a}{N^\alpha} + s.$$

To facilitate understanding, we conduct the derivation under the predefined fixed configuration: $G = 7$, $S = 0.31$ and $N = 1T$. The expression for $L^*(C)$ is provided as follows and illustrated in Figure 16:

$$L^*(C) \approx 1.87 + 576 \cdot C^{-0.158} \tag{33}$$

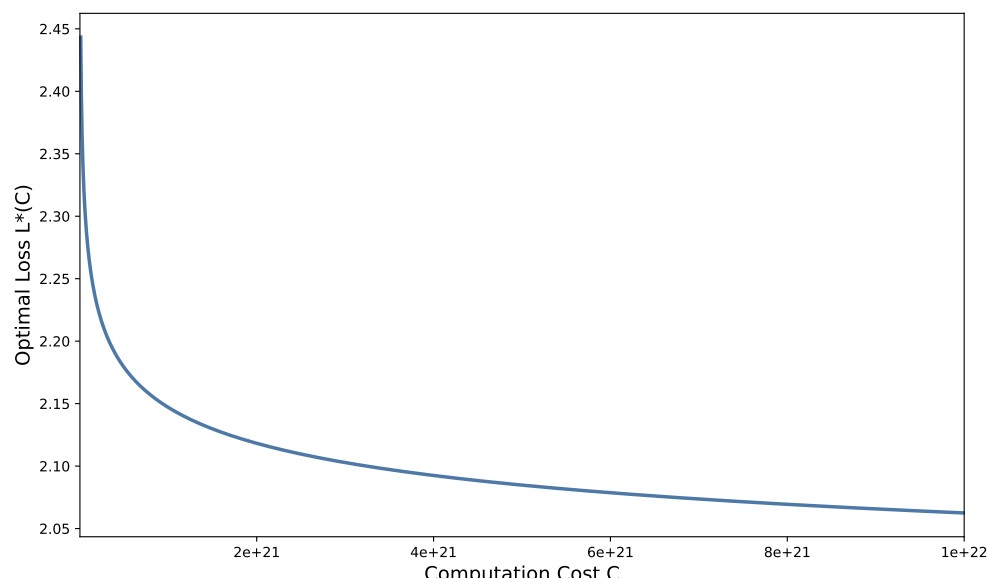

Figure 16: Illustration of $L^*(C)$ under the configuration of $N = 1\text{T}$, $G = 7$ and $S = 0.31$.

# H  DERIVATION OF THE $u$-$v$ RELATIONSHIP FOR $N_a$

To achieve the exclusive variation of the control variable $N_a$, it can be derived from Eq. 3 that when $G$ and $S$ are fixed, $N_a$ needs to be adjusted by modifying the expert dimension and the number of routed experts. Specifically, we scaled the expert dimension by a factor of $u$ while scaling down the number of routed experts by a corresponding factor $v$. However, to ensure that $N$ remains unchanged simultaneously, the following constraint applies:

$$\big(4d_{\text{head}} \cdot n_h + 3d_{\text{expert}}(SG + n_e)\big)d_{\text{hidden}} \cdot l = \big(4d_{\text{head}} \cdot n_h + 3d_{\text{expert}} \cdot u(SG + n_e \cdot v)\big)d_{\text{hidden}} \cdot l \quad (34)$$

Through formula transformation, we can derive:

$$v = \frac{(1 - u) \cdot S \cdot G + n_e}{u \cdot n_e}. \quad (35)$$

Thus, by setting the ratio of $u$ to $v$ according to the aforementioned Eq. 35, the controlled variation of $N_a$ can be achieved. For instance, when $N = 2.4\text{B}$, $G = 20$ and $S = 0.2$, $N_a$ takes values in the set $\{303\text{M}, 476\text{M}, 819\text{M}, 1507\text{M}, 2196\text{M}\}$, with the corresponding $d_{\text{expert}}$ being $\{112, 224, 448, 896, 1344\}$ and the respective $n_e$ being $\{260, 128, 62, 29, 18\}$.

# I  RELATED SCALING LAWS

Table 5: Comparison with existing MoE scaling laws.

|  | $N$ | $Na$ | $D$ | $G$ | $S$ |
| --- | --- | --- | --- | --- | --- |
| Scaling Laws for Fine-Grained MoE (Krajewski et al., 2024) | ✓ | ✗ | ✓ | ✓ | ✗ |
| Parameters vs. FLOPs (Abnar et al., 2025) | ✓ | ✓ | ✓ | ✗ | ✗ |
| Joint MoE Scaling Laws (Ludziejewski et al., 2025) | ✓ | ✓ | ✓ | ✗ | ✗ |
| Scaling Laws Across Architectures (Wang et al., 2024b) | ✓ | ✗ | ✓ | ✗ | ✗ |
| Unified Routed LMs (Clark et al., 2022) | ✓ | ✗ | ✗ | ✗ | ✗ |
| Our MoE Scaling Law | ✓ | ✓ | ✓ | ✓ | ✓ |

We selected two related scaling laws for comparison, each focusing on different aspects. Below, we briefly introduce their main ideas and describe how we use them to compare with the scaling law derived from our experiments.

## I.1 Scaling Laws for Fine-Grained Mixture-of-Experts

Krajewski et al. (2024) introduced a *granularity* factor into MoE scaling laws. They model the training loss $\mathcal{L}$ as a function of the number of **active parameters** $N$, the dataset size $D$ and the **granularity** $G$:

$$\mathcal{L}(N, D, G) = c + \left(\frac{g}{G^\gamma} + a\right)\frac{1}{N^\alpha} + \frac{b}{D^\beta}. \tag{36}$$

Here $c$ is the irreducible loss. The term $N^{-\alpha}$ captures the effect of model size, adjusted by $G$: finer experts (larger $G$) reduce this contribution. The last term $D^{-\beta}$ reflects the improvement from more data. Overall, the law indicates diminishing returns from $N$ and $D$, with better performance at higher $G$. As shown in Figure 5, this fitted result is less accurate than ours. The detailed fitted constants and values in Eq. 36 are presented below in Table 6.

Table 6: Fitted constants and their values in Eq. 36.

| Constant | Value |
|---|---|
| $a$ | 81.9404 |
| $\alpha$ | 0.2380 |
| $b$ | 3195.7723 |
| $\beta$ | 0.3695 |
| $c$ | 1.7709 |
| $g$ | 0.0004 |
| $\gamma$ | 0.0028 |

## I.2 Scaling Laws for Optimal Sparsity in MoE

Abnar et al. (2025) analyzed the effect of **sparsity** $S$, defined as the fraction of inactive experts. Their scaling law is:

$$L(N, D, S) = \frac{a}{N^\alpha} + \frac{b}{D^\beta} + \frac{c}{(1-S)^\lambda} + \frac{d}{(1-S)^\delta N^\gamma} + e. \tag{37}$$

The first two terms represent the standard effects of model and data size. The third term penalizes high sparsity, while the fourth couples sparsity with model size. The constant $e$ is an offset. This form shows that both the level of sparsity and its interaction with $N$ influence the loss. As shown in Figure 5, our method achieves a better fit than this result. The detailed fitted constants and values in Eq. 37 are presented below in Table 7.

Table 7: Fitted constants and their values in Eq. 37.

| Constant | Value |
|---|---|
| $a$ | 43.2889 |
| $\alpha$ | 0.1948 |
| $b$ | 8280.7176 |
| $\beta$ | 0.4145 |
| $e$ | 1.6351 |
| $c$ | 0.0003 |
| $d$ | 9.4982 |
| $\lambda$ | 1.8469 |
| $\delta$ | 0.0802 |
| $\gamma$ | 14.0591 |

## I.3 Comparison Methodology

For a fair comparison, we use the same dataset to fit as ours for these scaling laws. Each formula is applied without modification and parameters are optimized on identical experimental results under the same settings. This ensures that any differences arise solely from the functional forms. In

addition, we also fit other MoE scaling law (Ludziejewski et al., 2025; Clark et al., 2022), but their performance is likewise inferior to ours. This is likely because our formulation incorporates a more comprehensive set of MoE factors—$D, N, N_a, G, S$—and is applied over a broader range of MoE settings. Furthermore, our fitting carefully accounts for marginal effects while leveraging Occam's razor to simplify both hyperparameters and functional forms.

## J  LIMITATIONS

Our work has several limitations. In this work, we mainly focus on the classical MoE architecture. The analysis has not been validated at extremely larger scales or with alternative MoE architectures / training objectives due to the resource limit. In addition, we focus primarily on MoE-related factors and do not examine other components of LLMs that may also impact the performance of MoE, such as the attention layers and their interactions. Future work should extend the analysis to broader architectures and assess robustness.

## K  VARIATION OF DIFFERENT FACTORS

In our study, the core factors, including $N_a$, $N$, $G$, and $S$, are varied in strict adherence to the principle of controlling variables. We have already discussed the methods used to control these factors during the experiments in Appendix C. To clarify the experimental procedure, we provide a series of illustrations in Figure 17 that demonstrate how each variable is systematically adjusted:

## L  DETAILED SETTINGS OF EXPERIMENTS

We show the detailed configurations of our experiments as follows. Precisely, the value ranges of various factors in the experimental setup are as follows: for fitting data points, $G \in (1, 20]$, $S \in [0.0, 0.8]$, $N_a \in [30M, 2.2B]$, $N \in [133M, 3.4B]$ and $D \in [10B, 50B]$; for validation data points, $G \in (1, 40]$, $S \in [0.0, 0.7]$, $N_a \in [453M, 6.6B]$, $N \in [2.4B, 30B]$ and $D \in [10B, 100B]$.

We should highlight that our joint MoE scaling law is designed to accommodate practical experimental configurations. Therefore, the MoE configurations are more focused on the relatively practical settings, with partial of extreme settings to reveal the marginal effect of different factors. For the validation data, we adopt larger model/data sizes and broader ranges of factors to evaluate the effectiveness of our scaling law. For some uncommon and impractical experimental setups, such as those with extremely large $G$ and exceptionally high $N_a/N$ ratios simultaneously (whose efficiency is unsatisfactory in the view of MoE models), perfect fitting is not pursued by our MoE scaling law. The detailed configurations are given in Table 8.

Table 8: All configurations of experiments. The last column is **Label**, where ✓ indicates that the data point is used for validation and ✗ indicates that the data point is used for fitting or for observing marginal patterns.

|    | Na   | N    | D    | G    | S   | Label |    | Na   | N    | D    | G    | S   | Label |
|----|------|------|------|------|-----|-------|----|------|------|------|------|-----|-------|
| 0  | 48M  | 247M | 10B  | 2.5  | 0.2 | ✗     | 1  | 48M  | 247M | 10B  | 5    | 0.2 | ✗     |
| 2  | 48M  | 247M | 10B  | 10   | 0.2 | ✗     | 3  | 48M  | 247M | 10B  | 15   | 0.2 | ✗     |
| 4  | 48M  | 247M | 10B  | 20   | 0.2 | ✗     | 5  | 48M  | 247M | 20B  | 2.5  | 0.2 | ✗     |
| 6  | 48M  | 247M | 20B  | 5    | 0.2 | ✗     | 7  | 48M  | 247M | 20B  | 10   | 0.2 | ✗     |
| 8  | 48M  | 247M | 20B  | 15   | 0.2 | ✗     | 9  | 48M  | 247M | 20B  | 20   | 0.2 | ✗     |
| 10 | 48M  | 247M | 50B  | 2.5  | 0.2 | ✗     | 11 | 48M  | 247M | 50B  | 5    | 0.2 | ✗     |
| 12 | 48M  | 247M | 50B  | 10   | 0.2 | ✗     | 13 | 48M  | 247M | 50B  | 15   | 0.2 | ✗     |
| 14 | 48M  | 247M | 50B  | 20   | 0.2 | ✗     | 15 | 99M  | 496M | 10B  | 2.5  | 0.2 | ✗     |
| 16 | 99M  | 496M | 10B  | 5    | 0.2 | ✗     | 17 | 99M  | 496M | 10B  | 10   | 0.2 | ✗     |
| 18 | 99M  | 496M | 10B  | 15   | 0.2 | ✗     | 19 | 99M  | 496M | 10B  | 20   | 0.2 | ✗     |
| 20 | 99M  | 496M | 20B  | 2.5  | 0.2 | ✗     | 21 | 99M  | 496M | 20B  | 5    | 0.2 | ✗     |
| 22 | 99M  | 496M | 20B  | 10   | 0.2 | ✗     | 23 | 99M  | 496M | 20B  | 15   | 0.2 | ✗     |

**Table 8 continued from previous page**

| | Na | N | D | G | S | Label | | Na | N | D | G | S | Label |
|---|---|---|---|---|---|---|---|---|---|---|---|---|---|
| 24 | 99M | 496M | 20B | 20 | 0.2 | ✗ | 25 | 99M | 496M | 50B | 2.5 | 0.2 | ✗ |
| 26 | 99M | 496M | 50B | 5 | 0.2 | ✗ | 27 | 99M | 496M | 50B | 10 | 0.2 | ✗ |
| 28 | 99M | 496M | 50B | 15 | 0.2 | ✗ | 29 | 99M | 496M | 50B | 20 | 0.2 | ✗ |
| 30 | 181M | 907M | 10B | 2.5 | 0.2 | ✗ | 31 | 181M | 907M | 10B | 5 | 0.2 | ✗ |
| 32 | 181M | 907M | 10B | 10 | 0.2 | ✗ | 33 | 181M | 907M | 10B | 15 | 0.2 | ✗ |
| 34 | 181M | 907M | 10B | 20 | 0.2 | ✗ | 35 | 181M | 907M | 20B | 2.5 | 0.2 | ✗ |
| 36 | 181M | 907M | 20B | 5 | 0.2 | ✗ | 37 | 181M | 907M | 20B | 10 | 0.2 | ✗ |
| 38 | 181M | 907M | 20B | 15 | 0.2 | ✗ | 39 | 181M | 907M | 20B | 20 | 0.2 | ✗ |
| 40 | 181M | 907M | 50B | 2.5 | 0.2 | ✗ | 41 | 181M | 907M | 50B | 5 | 0.2 | ✗ |
| 42 | 181M | 907M | 50B | 10 | 0.2 | ✗ | 43 | 181M | 907M | 50B | 15 | 0.2 | ✗ |
| 44 | 181M | 907M | 50B | 20 | 0.2 | ✗ | 45 | 48M | 133M | 10B | 10 | 0.2 | ✗ |
| 46 | 48M | 247M | 10B | 10 | 0.2 | ✗ | 47 | 48M | 473M | 10B | 10 | 0.2 | ✗ |
| 48 | 48M | 926M | 10B | 10 | 0.2 | ✗ | 49 | 48M | 133M | 20B | 10 | 0.2 | ✗ |
| 50 | 48M | 247M | 20B | 10 | 0.2 | ✗ | 51 | 48M | 473M | 20B | 10 | 0.2 | ✗ |
| 52 | 48M | 926M | 20B | 10 | 0.2 | ✗ | 53 | 48M | 133M | 50B | 10 | 0.2 | ✗ |
| 54 | 48M | 247M | 50B | 10 | 0.2 | ✗ | 55 | 48M | 473M | 50B | 10 | 0.2 | ✗ |
| 56 | 48M | 926M | 50B | 10 | 0.2 | ✗ | 57 | 99M | 269M | 10B | 10 | 0.2 | ✗ |
| 58 | 99M | 496M | 10B | 10 | 0.2 | ✗ | 59 | 99M | 949M | 10B | 10 | 0.2 | ✗ |
| 60 | 99M | 1855M | 10B | 10 | 0.2 | ✗ | 61 | 99M | 269M | 20B | 10 | 0.2 | ✗ |
| 62 | 99M | 496M | 20B | 10 | 0.2 | ✗ | 63 | 99M | 949M | 20B | 10 | 0.2 | ✗ |
| 64 | 99M | 1855M | 20B | 10 | 0.2 | ✗ | 65 | 99M | 269M | 50B | 10 | 0.2 | ✗ |
| 66 | 99M | 496M | 50B | 10 | 0.2 | ✗ | 67 | 99M | 949M | 50B | 10 | 0.2 | ✗ |
| 68 | 99M | 1855M | 50B | 10 | 0.2 | ✗ | 69 | 181M | 492M | 10B | 10 | 0.2 | ✗ |
| 70 | 181M | 907M | 10B | 10 | 0.2 | ✗ | 71 | 181M | 1738M | 10B | 10 | 0.2 | ✗ |
| 72 | 181M | 3399M | 10B | 10 | 0.2 | ✗ | 73 | 181M | 492M | 20B | 10 | 0.2 | ✗ |
| 74 | 181M | 907M | 20B | 10 | 0.2 | ✗ | 75 | 181M | 1738M | 20B | 10 | 0.2 | ✗ |
| 76 | 181M | 3399M | 20B | 10 | 0.2 | ✗ | 77 | 181M | 492M | 50B | 10 | 0.2 | ✗ |
| 78 | 181M | 907M | 50B | 10 | 0.2 | ✗ | 79 | 181M | 1738M | 50B | 10 | 0.2 | ✗ |
| 80 | 181M | 3399M | 50B | 10 | 0.2 | ✗ | 81 | 48M | 247M | 10B | 10 | 0.0 | ✗ |
| 82 | 48M | 247M | 10B | 10 | 0.1 | ✗ | 83 | 48M | 247M | 10B | 10 | 0.2 | ✗ |
| 84 | 48M | 247M | 10B | 10 | 0.4 | ✗ | 85 | 48M | 247M | 20B | 10 | 0.0 | ✗ |
| 86 | 48M | 247M | 20B | 10 | 0.1 | ✗ | 87 | 48M | 247M | 20B | 10 | 0.2 | ✗ |
| 88 | 48M | 247M | 20B | 10 | 0.4 | ✗ | 89 | 48M | 247M | 50B | 10 | 0.0 | ✗ |
| 90 | 48M | 247M | 50B | 10 | 0.1 | ✗ | 91 | 48M | 247M | 50B | 10 | 0.2 | ✗ |
| 92 | 48M | 247M | 50B | 10 | 0.4 | ✗ | 93 | 99M | 496M | 10B | 10 | 0.0 | ✗ |
| 94 | 99M | 496M | 10B | 10 | 0.1 | ✗ | 95 | 99M | 496M | 10B | 10 | 0.2 | ✗ |
| 96 | 99M | 496M | 10B | 10 | 0.4 | ✗ | 97 | 99M | 496M | 20B | 10 | 0.0 | ✗ |
| 98 | 99M | 496M | 20B | 10 | 0.1 | ✗ | 99 | 99M | 496M | 20B | 10 | 0.2 | ✗ |
| 100 | 99M | 496M | 20B | 10 | 0.4 | ✗ | 101 | 99M | 496M | 50B | 10 | 0.0 | ✗ |
| 102 | 99M | 496M | 50B | 10 | 0.1 | ✗ | 103 | 99M | 496M | 50B | 10 | 0.2 | ✗ |
| 104 | 99M | 496M | 50B | 10 | 0.4 | ✗ | 105 | 181M | 907M | 10B | 10 | 0.0 | ✗ |
| 106 | 181M | 907M | 10B | 10 | 0.1 | ✗ | 107 | 181M | 907M | 10B | 10 | 0.2 | ✗ |
| 108 | 181M | 907M | 10B | 10 | 0.3 | ✗ | 109 | 181M | 907M | 10B | 10 | 0.4 | ✗ |
| 110 | 181M | 907M | 10B | 10 | 0.5 | ✗ | 111 | 181M | 907M | 10B | 10 | 0.6 | ✗ |
| 112 | 181M | 907M | 10B | 10 | 0.7 | ✗ | 113 | 181M | 907M | 20B | 10 | 0.0 | ✗ |
| 114 | 181M | 907M | 20B | 10 | 0.1 | ✗ | 115 | 181M | 907M | 20B | 10 | 0.2 | ✗ |
| 116 | 181M | 907M | 20B | 10 | 0.3 | ✗ | 117 | 181M | 907M | 20B | 10 | 0.4 | ✗ |
| 118 | 181M | 907M | 20B | 10 | 0.5 | ✗ | 119 | 181M | 907M | 20B | 10 | 0.6 | ✗ |
| 120 | 181M | 907M | 20B | 10 | 0.7 | ✗ | 121 | 181M | 907M | 50B | 10 | 0.0 | ✗ |
| 122 | 181M | 907M | 50B | 10 | 0.1 | ✗ | 123 | 181M | 907M | 50B | 10 | 0.2 | ✗ |
| 124 | 181M | 907M | 50B | 10 | 0.3 | ✗ | 125 | 181M | 907M | 50B | 10 | 0.4 | ✗ |
| 126 | 181M | 907M | 50B | 10 | 0.5 | ✗ | 127 | 181M | 907M | 50B | 10 | 0.6 | ✗ |
| 128 | 181M | 907M | 50B | 10 | 0.7 | ✗ | 129 | 240M | 1209M | 10B | 10 | 0.0 | ✗ |
| 130 | 240M | 1209M | 10B | 10 | 0.1 | ✗ | 131 | 240M | 1209M | 10B | 10 | 0.2 | ✗ |
| 132 | 240M | 1209M | 10B | 10 | 0.3 | ✗ | 133 | 240M | 1209M | 10B | 10 | 0.4 | ✗ |

**Table 8 continued from previous page**

| | Na | N | D | G | S | Label | | Na | N | D | G | S | Label |
|---|---|---|---|---|---|---|---|---|---|---|---|---|---|
| 134 | 240M | 1209M | 10B | 10 | 0.5 | ✗ | 135 | 240M | 1209M | 10B | 10 | 0.6 | ✗ |
| 136 | 240M | 1209M | 10B | 10 | 0.7 | ✗ | 137 | 240M | 1209M | 20B | 10 | 0.0 | ✗ |
| 138 | 240M | 1209M | 20B | 10 | 0.1 | ✗ | 139 | 240M | 1209M | 20B | 10 | 0.2 | ✗ |
| 140 | 240M | 1209M | 20B | 10 | 0.3 | ✗ | 141 | 240M | 1209M | 20B | 10 | 0.4 | ✗ |
| 142 | 240M | 1209M | 20B | 10 | 0.5 | ✗ | 143 | 240M | 1209M | 20B | 10 | 0.6 | ✗ |
| 144 | 240M | 1209M | 20B | 10 | 0.7 | ✗ | 145 | 240M | 1209M | 50B | 10 | 0.0 | ✗ |
| 146 | 240M | 1209M | 50B | 10 | 0.1 | ✗ | 147 | 240M | 1209M | 50B | 10 | 0.2 | ✗ |
| 148 | 240M | 1209M | 50B | 10 | 0.3 | ✗ | 149 | 240M | 1209M | 50B | 10 | 0.4 | ✗ |
| 150 | 240M | 1209M | 50B | 10 | 0.5 | ✗ | 151 | 240M | 1209M | 50B | 10 | 0.6 | ✗ |
| 152 | 240M | 1209M | 50B | 10 | 0.7 | ✗ | 153 | 476M | 2404M | 10B | 10 | 0.0 | ✗ |
| 154 | 476M | 2404M | 10B | 10 | 0.1 | ✗ | 155 | 476M | 2404M | 10B | 10 | 0.3 | ✗ |
| 156 | 476M | 2404M | 10B | 10 | 0.4 | ✗ | 157 | 476M | 2404M | 10B | 10 | 0.5 | ✗ |
| 158 | 476M | 2404M | 10B | 10 | 0.6 | ✗ | 159 | 476M | 2404M | 10B | 10 | 0.7 | ✗ |
| 160 | 476M | 2404M | 20B | 10 | 0.0 | ✗ | 161 | 476M | 2404M | 20B | 10 | 0.1 | ✗ |
| 162 | 476M | 2404M | 20B | 10 | 0.3 | ✗ | 163 | 476M | 2404M | 20B | 10 | 0.4 | ✗ |
| 164 | 476M | 2404M | 20B | 10 | 0.5 | ✗ | 165 | 476M | 2404M | 20B | 10 | 0.6 | ✗ |
| 166 | 476M | 2404M | 20B | 10 | 0.7 | ✗ | 167 | 476M | 2404M | 50B | 10 | 0.0 | ✗ |
| 168 | 476M | 2404M | 50B | 10 | 0.1 | ✗ | 169 | 476M | 2404M | 50B | 10 | 0.3 | ✗ |
| 170 | 476M | 2404M | 50B | 10 | 0.4 | ✗ | 171 | 476M | 2404M | 50B | 10 | 0.5 | ✗ |
| 172 | 476M | 2404M | 50B | 10 | 0.6 | ✗ | 173 | 476M | 2404M | 50B | 10 | 0.7 | ✗ |
| 174 | 240M | 1209M | 10B | 5 | 0.0 | ✗ | 175 | 240M | 1209M | 10B | 5 | 0.2 | ✗ |
| 176 | 240M | 1209M | 10B | 5 | 0.4 | ✗ | 177 | 240M | 1209M | 10B | 5 | 0.6 | ✗ |
| 178 | 240M | 1209M | 10B | 5 | 0.8 | ✗ | 179 | 240M | 1209M | 20B | 5 | 0.0 | ✗ |
| 180 | 240M | 1209M | 20B | 5 | 0.2 | ✗ | 181 | 240M | 1209M | 20B | 5 | 0.4 | ✗ |
| 182 | 240M | 1209M | 20B | 5 | 0.6 | ✗ | 183 | 240M | 1209M | 20B | 5 | 0.8 | ✗ |
| 184 | 240M | 1209M | 50B | 5 | 0.0 | ✗ | 185 | 240M | 1209M | 50B | 5 | 0.2 | ✗ |
| 186 | 240M | 1209M | 50B | 5 | 0.4 | ✗ | 187 | 240M | 1209M | 50B | 5 | 0.6 | ✗ |
| 188 | 240M | 1209M | 50B | 5 | 0.8 | ✗ | 189 | 476M | 2404M | 10B | 5 | 0.0 | ✗ |
| 190 | 476M | 2404M | 10B | 5 | 0.2 | ✗ | 191 | 476M | 2404M | 10B | 5 | 0.4 | ✗ |
| 192 | 476M | 2404M | 10B | 5 | 0.6 | ✗ | 193 | 476M | 2404M | 10B | 5 | 0.8 | ✗ |
| 194 | 476M | 2404M | 20B | 5 | 0.0 | ✗ | 195 | 476M | 2404M | 20B | 5 | 0.2 | ✗ |
| 196 | 476M | 2404M | 20B | 5 | 0.4 | ✗ | 197 | 476M | 2404M | 20B | 5 | 0.6 | ✗ |
| 198 | 476M | 2404M | 20B | 5 | 0.8 | ✗ | 199 | 476M | 2404M | 50B | 5 | 0.0 | ✗ |
| 200 | 476M | 2404M | 50B | 5 | 0.2 | ✗ | 201 | 476M | 2404M | 50B | 5 | 0.4 | ✗ |
| 202 | 476M | 2404M | 50B | 5 | 0.6 | ✗ | 203 | 476M | 2404M | 50B | 5 | 0.8 | ✗ |
| 204 | 31M | 247M | 10B | 20 | 0.2 | ✗ | 205 | 31M | 247M | 20B | 20 | 0.2 | ✗ |
| 206 | 31M | 247M | 30B | 20 | 0.2 | ✗ | 207 | 31M | 247M | 50B | 20 | 0.2 | ✗ |
| 208 | 64M | 496M | 10B | 20 | 0.2 | ✗ | 209 | 99M | 496M | 10B | 20 | 0.2 | ✗ |
| 210 | 170M | 496M | 10B | 20 | 0.2 | ✗ | 211 | 312M | 496M | 10B | 20 | 0.2 | ✗ |
| 212 | 453M | 496M | 10B | 20 | 0.2 | ✗ | 213 | 64M | 496M | 20B | 20 | 0.2 | ✗ |
| 214 | 99M | 496M | 20B | 20 | 0.2 | ✗ | 215 | 170M | 496M | 20B | 20 | 0.2 | ✗ |
| 216 | 312M | 496M | 20B | 20 | 0.2 | ✗ | 217 | 453M | 496M | 20B | 20 | 0.2 | ✗ |
| 218 | 64M | 496M | 30B | 20 | 0.2 | ✗ | 219 | 99M | 496M | 30B | 20 | 0.2 | ✗ |
| 220 | 170M | 496M | 30B | 20 | 0.2 | ✗ | 221 | 312M | 496M | 30B | 20 | 0.2 | ✗ |
| 222 | 453M | 496M | 30B | 20 | 0.2 | ✗ | 223 | 64M | 496M | 50B | 20 | 0.2 | ✗ |
| 224 | 99M | 496M | 50B | 20 | 0.2 | ✗ | 225 | 170M | 496M | 50B | 20 | 0.2 | ✗ |
| 226 | 312M | 496M | 50B | 20 | 0.2 | ✗ | 227 | 453M | 496M | 50B | 20 | 0.2 | ✗ |
| 228 | 116M | 907M | 10B | 20 | 0.2 | ✗ | 229 | 181M | 907M | 10B | 20 | 0.2 | ✗ |
| 230 | 310M | 907M | 10B | 20 | 0.2 | ✗ | 231 | 570M | 907M | 10B | 20 | 0.2 | ✗ |
| 232 | 829M | 907M | 10B | 20 | 0.2 | ✗ | 233 | 116M | 907M | 20B | 20 | 0.2 | ✗ |
| 234 | 181M | 907M | 20B | 20 | 0.2 | ✗ | 235 | 310M | 907M | 20B | 20 | 0.2 | ✗ |
| 236 | 570M | 907M | 20B | 20 | 0.2 | ✗ | 237 | 829M | 907M | 20B | 20 | 0.2 | ✗ |
| 238 | 116M | 907M | 30B | 20 | 0.2 | ✗ | 239 | 181M | 907M | 30B | 20 | 0.2 | ✗ |
| 240 | 310M | 907M | 30B | 20 | 0.2 | ✗ | 241 | 570M | 907M | 30B | 20 | 0.2 | ✗ |
| 242 | 829M | 907M | 30B | 20 | 0.2 | ✗ | 243 | 116M | 907M | 50B | 20 | 0.2 | ✗ |

**Table 8 continued from previous page**

| | Na | N | D | G | S | Label | | Na | N | D | G | S | Label |
|---|---|---|---|---|---|---|---|---|---|---|---|---|---|
| 244 | 181M | 907M | 50B | 20 | 0.2 | ✗ | 245 | 310M | 907M | 50B | 20 | 0.2 | ✗ |
| 246 | 570M | 907M | 50B | 20 | 0.2 | ✗ | 247 | 829M | 907M | 50B | 20 | 0.2 | ✗ |
| 248 | 304M | 2404M | 10B | 20 | 0.2 | ✗ | 249 | 476M | 2404M | 10B | 20 | 0.2 | ✗ |
| 250 | 820M | 2404M | 10B | 20 | 0.2 | ✗ | 251 | 1508M | 2404M | 10B | 20 | 0.2 | ✗ |
| 252 | 2196M | 2404M | 10B | 20 | 0.2 | ✗ | 253 | 304M | 2404M | 20B | 20 | 0.2 | ✗ |
| 254 | 476M | 2404M | 20B | 20 | 0.2 | ✗ | 255 | 820M | 2404M | 20B | 20 | 0.2 | ✗ |
| 256 | 1508M | 2404M | 20B | 20 | 0.2 | ✗ | 257 | 2196M | 2404M | 20B | 20 | 0.2 | ✗ |
| 258 | 304M | 2404M | 30B | 20 | 0.2 | ✗ | 259 | 476M | 2404M | 30B | 20 | 0.2 | ✗ |
| 260 | 820M | 2404M | 30B | 20 | 0.2 | ✗ | 261 | 1508M | 2404M | 30B | 20 | 0.2 | ✗ |
| 262 | 2196M | 2404M | 30B | 20 | 0.2 | ✗ | 263 | 304M | 2404M | 50B | 20 | 0.2 | ✗ |
| 264 | 476M | 2404M | 50B | 20 | 0.2 | ✗ | 265 | 820M | 2404M | 50B | 20 | 0.2 | ✗ |
| 266 | 1508M | 2404M | 50B | 20 | 0.2 | ✗ | 267 | 2196M | 2404M | 50B | 20 | 0.2 | ✗ |
| 268 | 22M | 121M | 10B | 1 | 0.0 | ✗ | 269 | 22M | 121M | 10B | 2 | 0.0 | ✗ |
| 270 | 22M | 121M | 10B | 3 | 0.0 | ✗ | 271 | 22M | 121M | 10B | 4 | 0.0 | ✗ |
| 272 | 22M | 121M | 10B | 8 | 0.0 | ✗ | 273 | 22M | 121M | 10B | 16 | 0.0 | ✗ |
| 274 | 22M | 121M | 20B | 1 | 0.0 | ✗ | 275 | 22M | 121M | 20B | 2 | 0.0 | ✗ |
| 276 | 22M | 121M | 20B | 3 | 0.0 | ✗ | 277 | 22M | 121M | 20B | 4 | 0.0 | ✗ |
| 278 | 22M | 121M | 20B | 8 | 0.0 | ✗ | 279 | 22M | 121M | 20B | 16 | 0.0 | ✗ |
| 280 | 22M | 121M | 50B | 1 | 0.0 | ✗ | 281 | 22M | 121M | 50B | 2 | 0.0 | ✗ |
| 282 | 22M | 121M | 50B | 3 | 0.0 | ✗ | 283 | 22M | 121M | 50B | 4 | 0.0 | ✗ |
| 284 | 22M | 121M | 50B | 8 | 0.0 | ✗ | 285 | 22M | 121M | 50B | 16 | 0.0 | ✗ |
| 286 | 30M | 93M | 10B | 1 | 0.0 | ✗ | 287 | 30M | 93M | 10B | 2 | 0.0 | ✗ |
| 288 | 30M | 93M | 10B | 3 | 0.0 | ✗ | 289 | 30M | 93M | 10B | 4 | 0.0 | ✗ |
| 290 | 30M | 93M | 10B | 8 | 0.0 | ✗ | 291 | 30M | 93M | 10B | 16 | 0.0 | ✗ |
| 292 | 30M | 93M | 20B | 1 | 0.0 | ✗ | 293 | 30M | 93M | 20B | 2 | 0.0 | ✗ |
| 294 | 30M | 93M | 20B | 3 | 0.0 | ✗ | 295 | 30M | 93M | 20B | 4 | 0.0 | ✗ |
| 296 | 30M | 93M | 20B | 8 | 0.0 | ✗ | 297 | 30M | 93M | 20B | 16 | 0.0 | ✗ |
| 298 | 30M | 93M | 50B | 1 | 0.0 | ✗ | 299 | 30M | 93M | 50B | 2 | 0.0 | ✗ |
| 300 | 30M | 93M | 50B | 3 | 0.0 | ✗ | 301 | 30M | 93M | 50B | 4 | 0.0 | ✗ |
| 302 | 30M | 93M | 50B | 8 | 0.0 | ✗ | 303 | 30M | 93M | 50B | 16 | 0.0 | ✗ |
| 304 | 30M | 175M | 10B | 1 | 0.0 | ✗ | 305 | 30M | 175M | 10B | 2 | 0.0 | ✗ |
| 306 | 30M | 175M | 10B | 3 | 0.0 | ✗ | 307 | 30M | 175M | 10B | 4 | 0.0 | ✗ |
| 308 | 30M | 175M | 10B | 8 | 0.0 | ✗ | 309 | 30M | 175M | 10B | 16 | 0.0 | ✗ |
| 310 | 30M | 175M | 20B | 1 | 0.0 | ✗ | 311 | 30M | 175M | 20B | 2 | 0.0 | ✗ |
| 312 | 30M | 175M | 20B | 3 | 0.0 | ✗ | 313 | 30M | 175M | 20B | 4 | 0.0 | ✗ |
| 314 | 30M | 175M | 20B | 8 | 0.0 | ✗ | 315 | 30M | 175M | 20B | 16 | 0.0 | ✗ |
| 316 | 30M | 175M | 50B | 1 | 0.0 | ✗ | 317 | 30M | 175M | 50B | 2 | 0.0 | ✗ |
| 318 | 30M | 175M | 50B | 3 | 0.0 | ✗ | 319 | 30M | 175M | 50B | 4 | 0.0 | ✗ |
| 320 | 30M | 175M | 50B | 8 | 0.0 | ✗ | 321 | 30M | 175M | 50B | 16 | 0.0 | ✗ |
| 322 | 30M | 340M | 10B | 1 | 0.0 | ✗ | 323 | 30M | 340M | 10B | 2 | 0.0 | ✗ |
| 324 | 30M | 340M | 10B | 3 | 0.0 | ✗ | 325 | 30M | 340M | 10B | 4 | 0.0 | ✗ |
| 326 | 30M | 340M | 10B | 8 | 0.0 | ✗ | 327 | 30M | 340M | 10B | 16 | 0.0 | ✗ |
| 328 | 30M | 340M | 20B | 1 | 0.0 | ✗ | 329 | 30M | 340M | 20B | 2 | 0.0 | ✗ |
| 330 | 30M | 340M | 20B | 3 | 0.0 | ✗ | 331 | 30M | 340M | 20B | 4 | 0.0 | ✗ |
| 332 | 30M | 340M | 20B | 8 | 0.0 | ✗ | 333 | 30M | 340M | 20B | 16 | 0.0 | ✗ |
| 334 | 30M | 340M | 50B | 1 | 0.0 | ✗ | 335 | 30M | 340M | 50B | 2 | 0.0 | ✗ |
| 336 | 30M | 340M | 50B | 3 | 0.0 | ✗ | 337 | 30M | 340M | 50B | 4 | 0.0 | ✗ |
| 338 | 30M | 340M | 50B | 8 | 0.0 | ✗ | 339 | 30M | 340M | 50B | 16 | 0.0 | ✗ |
| 340 | 41M | 239M | 10B | 1 | 0.0 | ✗ | 341 | 41M | 239M | 10B | 2 | 0.0 | ✗ |
| 342 | 41M | 239M | 10B | 3 | 0.0 | ✗ | 343 | 41M | 239M | 10B | 4 | 0.0 | ✗ |
| 344 | 41M | 239M | 10B | 8 | 0.0 | ✗ | 345 | 41M | 239M | 10B | 16 | 0.0 | ✗ |
| 346 | 41M | 239M | 20B | 1 | 0.0 | ✗ | 347 | 41M | 239M | 20B | 2 | 0.0 | ✗ |
| 348 | 41M | 239M | 20B | 3 | 0.0 | ✗ | 349 | 41M | 239M | 20B | 4 | 0.0 | ✗ |
| 350 | 41M | 239M | 20B | 8 | 0.0 | ✗ | 351 | 41M | 239M | 20B | 16 | 0.0 | ✗ |
| 352 | 41M | 239M | 50B | 1 | 0.0 | ✗ | 353 | 41M | 239M | 50B | 2 | 0.0 | ✗ |

**Table 8 continued from previous page**

| | Na | N | D | G | S | Label | | Na | N | D | G | S | Label |
|---|---|---|---|---|---|---|---|---|---|---|---|---|---|
| 354 | 41M | 239M | 50B | 3 | 0.0 | ✗ | 355 | 41M | 239M | 50B | 4 | 0.0 | ✗ |
| 356 | 41M | 239M | 50B | 8 | 0.0 | ✗ | 357 | 41M | 239M | 50B | 16 | 0.0 | ✗ |
| 358 | 476M | 1301M | 10B | 10 | 0.2 | ✓ | 359 | 476M | 2404M | 10B | 10 | 0.2 | ✓ |
| 360 | 476M | 4604M | 10B | 10 | 0.2 | ✓ | 361 | 476M | 9008M | 10B | 10 | 0.2 | ✓ |
| 362 | 476M | 1301M | 20B | 10 | 0.2 | ✓ | 363 | 476M | 2404M | 20B | 10 | 0.2 | ✓ |
| 364 | 476M | 4604M | 20B | 10 | 0.2 | ✓ | 365 | 476M | 9008M | 20B | 10 | 0.2 | ✓ |
| 366 | 476M | 1301M | 50B | 10 | 0.2 | ✓ | 367 | 476M | 2404M | 50B | 10 | 0.2 | ✓ |
| 368 | 476M | 4604M | 50B | 10 | 0.2 | ✓ | 369 | 476M | 9008M | 50B | 10 | 0.2 | ✓ |
| 370 | 793M | 3964M | 10B | 10 | 0.0 | ✓ | 371 | 793M | 3964M | 10B | 10 | 0.1 | ✓ |
| 372 | 793M | 3964M | 10B | 10 | 0.2 | ✓ | 373 | 793M | 3964M | 10B | 10 | 0.3 | ✓ |
| 374 | 793M | 3964M | 10B | 10 | 0.4 | ✓ | 375 | 793M | 3964M | 10B | 10 | 0.5 | ✓ |
| 376 | 793M | 3964M | 10B | 10 | 0.6 | ✓ | 377 | 793M | 3964M | 10B | 10 | 0.7 | ✓ |
| 378 | 793M | 3964M | 20B | 10 | 0.0 | ✓ | 379 | 793M | 3964M | 20B | 10 | 0.1 | ✓ |
| 380 | 793M | 3964M | 20B | 10 | 0.2 | ✓ | 381 | 793M | 3964M | 20B | 10 | 0.3 | ✓ |
| 382 | 793M | 3964M | 20B | 10 | 0.4 | ✓ | 383 | 793M | 3964M | 20B | 10 | 0.5 | ✓ |
| 384 | 793M | 3964M | 20B | 10 | 0.6 | ✓ | 385 | 793M | 3964M | 20B | 10 | 0.7 | ✓ |
| 386 | 793M | 3964M | 50B | 10 | 0.0 | ✓ | 387 | 793M | 3964M | 50B | 10 | 0.1 | ✓ |
| 388 | 793M | 3964M | 50B | 10 | 0.2 | ✓ | 389 | 793M | 3964M | 50B | 10 | 0.3 | ✓ |
| 390 | 793M | 3964M | 50B | 10 | 0.4 | ✓ | 391 | 793M | 3964M | 50B | 10 | 0.5 | ✓ |
| 392 | 793M | 3964M | 50B | 10 | 0.6 | ✓ | 393 | 793M | 3964M | 50B | 10 | 0.7 | ✓ |
| 394 | 1441M | 7255M | 10B | 10 | 0.2 | ✓ | 395 | 1960M | 7255M | 10B | 10 | 0.2 | ✓ |
| 396 | 2479M | 7255M | 10B | 10 | 0.2 | ✓ | 397 | 3517M | 7255M | 10B | 10 | 0.2 | ✓ |
| 398 | 6632M | 7255M | 10B | 10 | 0.2 | ✓ | 399 | 1441M | 7255M | 20B | 10 | 0.2 | ✓ |
| 400 | 1960M | 7255M | 20B | 10 | 0.2 | ✓ | 401 | 2479M | 7255M | 20B | 10 | 0.2 | ✓ |
| 402 | 3517M | 7255M | 20B | 10 | 0.2 | ✓ | 403 | 6632M | 7255M | 20B | 10 | 0.2 | ✓ |
| 404 | 1441M | 7255M | 50B | 10 | 0.2 | ✓ | 405 | 1960M | 7255M | 50B | 10 | 0.2 | ✓ |
| 406 | 2479M | 7255M | 50B | 10 | 0.2 | ✓ | 407 | 3517M | 7255M | 50B | 10 | 0.2 | ✓ |
| 408 | 6632M | 7255M | 50B | 10 | 0.2 | ✓ | 409 | 453M | 3964M | 10B | 2 | 0.5 | ✓ |
| 410 | 453M | 3964M | 10B | 4 | 0.5 | ✓ | 411 | 453M | 3964M | 10B | 6 | 0.5 | ✓ |
| 412 | 453M | 3964M | 10B | 8 | 0.5 | ✓ | 413 | 453M | 3964M | 10B | 10 | 0.5 | ✓ |
| 414 | 453M | 3964M | 10B | 12 | 0.5 | ✓ | 415 | 453M | 3964M | 20B | 2 | 0.5 | ✓ |
| 416 | 453M | 3964M | 20B | 4 | 0.5 | ✓ | 417 | 453M | 3964M | 20B | 6 | 0.5 | ✓ |
| 418 | 453M | 3964M | 20B | 8 | 0.5 | ✓ | 419 | 453M | 3964M | 20B | 10 | 0.5 | ✓ |
| 420 | 453M | 3964M | 20B | 12 | 0.5 | ✓ | 421 | 453M | 3964M | 50B | 2 | 0.5 | ✓ |
| 422 | 453M | 3964M | 50B | 4 | 0.5 | ✓ | 423 | 453M | 3964M | 50B | 6 | 0.5 | ✓ |
| 424 | 453M | 3964M | 50B | 8 | 0.5 | ✓ | 425 | 453M | 3964M | 50B | 10 | 0.5 | ✓ |
| 426 | 453M | 3964M | 50B | 12 | 0.5 | ✓ | 427 | 566M | 3964M | 10B | 3 | 0.33 | ✓ |
| 428 | 566M | 3964M | 10B | 6 | 0.33 | ✓ | 429 | 566M | 3964M | 10B | 9 | 0.33 | ✓ |
| 430 | 566M | 3964M | 10B | 12 | 0.33 | ✓ | 431 | 566M | 3964M | 20B | 3 | 0.33 | ✓ |
| 432 | 566M | 3964M | 20B | 6 | 0.33 | ✓ | 433 | 566M | 3964M | 20B | 9 | 0.33 | ✓ |
| 434 | 566M | 3964M | 20B | 12 | 0.33 | ✓ | 435 | 566M | 3964M | 50B | 3 | 0.33 | ✓ |
| 436 | 566M | 3964M | 50B | 6 | 0.33 | ✓ | 437 | 566M | 3964M | 50B | 9 | 0.33 | ✓ |
| 438 | 566M | 3964M | 50B | 12 | 0.33 | ✓ | 439 | 476M | 2404M | 10B | 2.5 | 0.2 | ✓ |
| 440 | 476M | 2404M | 20B | 2.5 | 0.2 | ✓ | 441 | 476M | 2404M | 50B | 2.5 | 0.2 | ✓ |
| 442 | 793M | 3964M | 100B | 10 | 0.2 | ✓ | 443 | 476M | 2404M | 10B | 20 | 0.2 | ✓ |
| 444 | 476M | 2404M | 20B | 20 | 0.2 | ✓ | 445 | 476M | 2404M | 50B | 20 | 0.2 | ✓ |
| 446 | 3070M | 30249M | 50B | 20 | 0.2 | ✓ | 447 | 181M | 907M | 10B | 40 | 0.2 | ✓ |
| 448 | 181M | 907M | 20B | 40 | 0.2 | ✓ | 449 | 181M | 907M | 50B | 40 | 0.2 | ✓ |

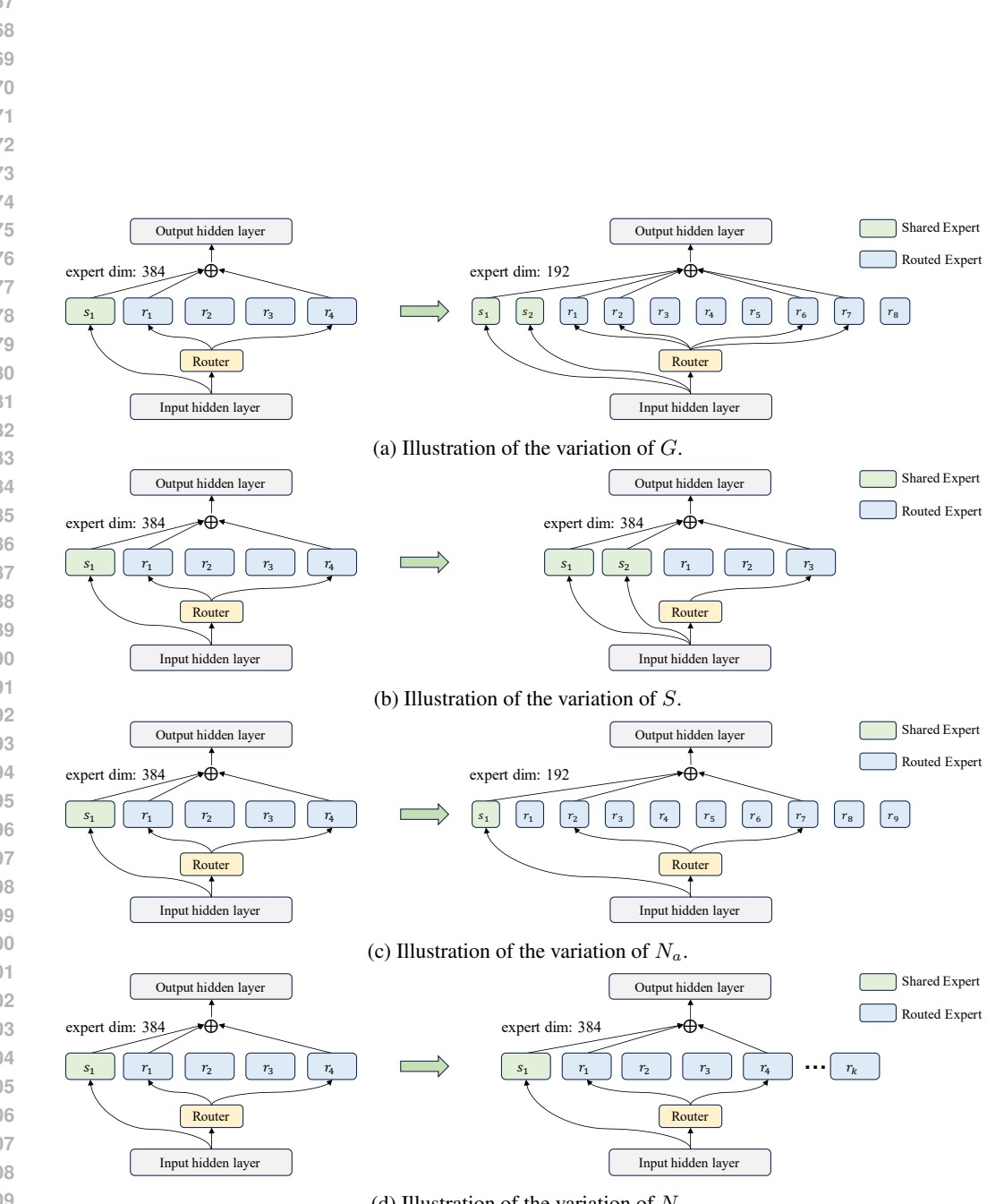

(a) Illustration of the variation of $G$.

(b) Illustration of the variation of $S$.

(c) Illustration of the variation of $N_a$.

(d) Illustration of the variation of $N$.

Figure 17: Illustration of the variation of factor $G$, $S$, $N_a$ and $N$ under controlled conditions.

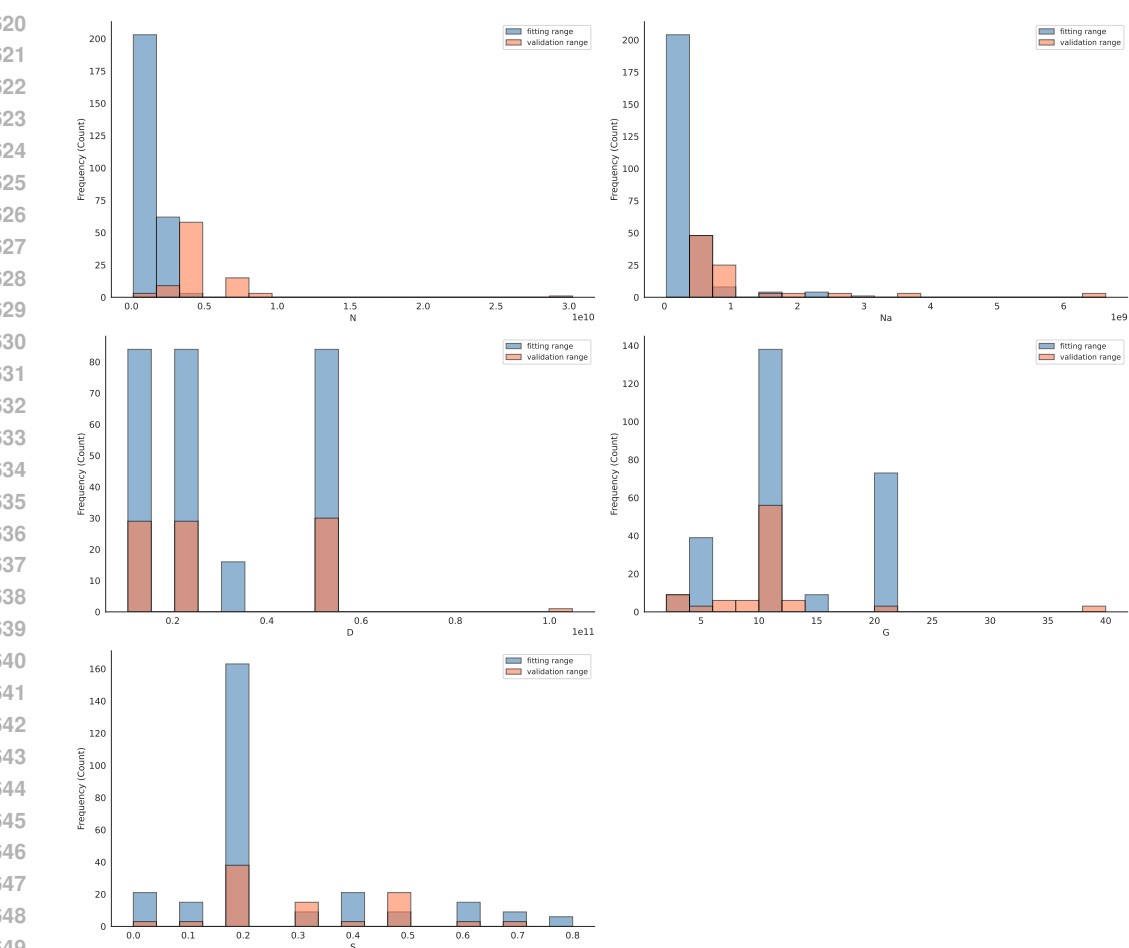

Figure 18: Distribution histograms of $N$, $N_a$, $D$, $G$, and $S$ for fitting data and validation data.

