# OpenReview forum: "Towards a Comprehensive Scaling Law of Mixture-of-Experts"
_ICLR.cc/2026/Conference — Submitted to ICLR 2026_

### Official Review · Reviewer_f3c7 · 2025-10-26

**Soundness:** 3
**Presentation:** 3
**Contribution:** 3
**Rating:** 6
**Confidence:** 5

**Summary:**

The study extends existing Mixture-of-Experts (MoE) scaling law research to encompass the architectural choices present in current frontier open models. It decomposes MoE configurations into five variables influencing performance: data size $(D)$, total model size $(N)$, activated model size $(N_a)$, number of activated experts $(G)$, and ratio of shared experts $(S)$ in the total activated experts.
Using $446$ controlled experiments, the authors quantify the marginal effects of these factors and construct a joint scaling law incorporating all five.
From the fitted scaling laws, analytical expressions are derived for the theoretically optimal and efficiency-aware values of $G$, $S$, and $N_a/N$, showing that the optimal $G$ and $S$ remain invariant with respect to architecture and data scale, while the optimal activation ratio $N_a/N$ decreases as total model size increases.
These optimal configurations are compared against existing frontier open MoE language models.
The resulting formulation provides a quantitative framework for predicting MoE performance and configuring model sparsity.

This work is more directly applicable for analysis and design of modern MoE LLMs, seeing as it directly deals with the number of activated experts or the number of shared experts. It also compares directly to existing open MoEs, and shows partial agreement between their predicted optimal configurations and choices made in open models - and where there is disagreement, elaborates on why that might be.

The hypothesis space considered in this work is larger than in the existing work, and as such can directly compare the quality of fit with other existing works by reproducing them on the same data - although I have reservations as to whether the comparisons are useful and what the takeaways are (more in Weaknesses/Questions).

**Strengths:**

1. **Wide scope** — compared to existing MoE scaling law studies, the formulation captures nearly all architectural and scaling choices used in frontier MoE LLMs.

2. **Progressive hypothesis construction** — development proceeds systematically from dense scaling laws (Chinchilla; Hoffmann et al., 2022) to a joint law over all five variables $(N, D, N_a, G, S)$, with validation at each stage.

3. **Comprehensive experimental design** — 446 controlled experiments are conducted to isolate and quantify the marginal effects of each factor.

4. **Analytical interpretation** — theoretical expressions for the optimal $G$, $S$, and $N_a/N$ are derived from the fitted law and compared with contemporary large MoE models.

5. **Transparent reporting** — all experimental configurations are listed, though the absence of of final loss values, as well as the fitting procedure, greatly hinders reproducibility.

**Weaknesses:**

I have several significant concerns that currently prevent me from recommending acceptance. I would welcome the authors' responses to these points, as they could substantially affect my evaluation.

### 1. Clarity needed on the fitting procedure

The description of the fitting methodology in the paper requires clarification. The current text states:

>To determine the specific value of hyperparameters in Eq. 11, we implement all 446 experiments that encompass diverse configurations of N, D, Na, G and S. The corresponding hyperparameter values are provided in Table 2 in Appendix C.1. Next, we evaluate the fitting performance of our MoE scaling law in Figure 5, where the satisfactory fitting performance demonstrates the advantage of our MoE scaling law compared to others. Notably, for the sake of fairness, during the comparison, the baseline MoE scaling laws were first re-fitted with hyperparameters using the same data points, followed by prediction.

This paragraph is difficult to comprehend. I would appreciate more detailed information on:
- The specific fitting algorithm or optimization method used
- How the validation set was utilized in the fitting process
- The meaning of "baseline MoE scaling laws were first re-fitted with hyperparameters using the same data points, followed by prediction"

My main concern is whether the validation set was used for final model selection. Given that the proposed model has 12 parameters (compared to 7 in Ludziejewski et al. (2024), 10 in Abnar et al. (2025), and 9 in Ludziejewski et al. (2025)), if the training set was used to generate candidate fits and the validation set was used for selection, models with more parameters may have an inherent advantage due to increased flexibility.

Additionally, while mean RMSE and visual comparisons are provided, I suggest including additional evaluation metrics, as recommended in recent work such as Besiroglu et al. (2024).

### 2. Clarification needed on validation set characteristics

I would appreciate explicit information about whether the validation set represents interpolation or extrapolation relative to the training datapoints. This distinction is important for interpreting the results. While this is possible to do based just on the appendix, it requires actual data analysis.

### 3. Suggestion for distributional comparison with prior work

The current comparison evaluates prior scaling laws on data that may be out-of-distribution for those functional forms. To strengthen the paper, I suggest including comparisons using data closer to the original experimental distributions of each prior work (Ludziejewski et al., 2024; Abnar et al., 2025; Ludziejewski et al., 2025). This would provide a fairer assessment of whether the proposed hypothesis space offers genuine improvements.

### 4. Minor concern regarding experimental controls

Following Porian et al. (2024), I note that the experimental design does not appear to adjust for batch size or learning rate across different scales, which could introduce bias, particularly at smaller scales. While this is a minor point, addressing it would strengthen the experimental foundation.

---

### References

- Kaplan, J., et al. (2020). [Scaling Laws for Neural Language Models](https://arxiv.org/abs/2001.08361).

- Hoffmann, J., et al. (2022). [Training compute-optimal large language models](https://proceedings.neurips.cc/paper_files/paper/2022/file/c1e2faff6f588870935f114ebe04a3e5-Paper-Conference.pdf). NeurIPS 2022.

- Ludziejewski, J., et al. (2024). [Scaling Laws for Fine-Grained Mixture of Experts](https://proceedings.mlr.press/v235/ludziejewski24a.html). ICML 2024.

- Porian, T., et al. (2024). [Resolving Discrepancies in Compute-Optimal Scaling of Language Models](https://proceedings.neurips.cc/paper_files/paper/2024/file/b6341525cd84f3be0ef203e4d7cd8556-Paper-Conference.pdf). NeurIPS 2024.

- Besiroglu, T., et al. (2024). [Chinchilla Scaling: A replication attempt](https://arxiv.org/abs/2404.10102).

- Abnar, S., et al. (2025). [Parameters vs FLOPs: Scaling Laws for Optimal Sparsity for Mixture-of-Experts Language Models](https://arxiv.org/abs/2501.12370). ICML 2025.

- Ludziejewski, J., et al. (2025). [Joint MoE Scaling Laws: Mixture of Experts Can Be Memory Efficient](https://proceedings.mlr.press/v267/ludziejewski25a.html). ICML 2025.

**Questions:**

### 1. Please describe the fitting procedure in detail.

The strongest description here would be: release the code + your data in an anonymous repo, together with fitting of other existing scaling laws. It would mean that you provide a script that starts with raw data, define the search grid (which I assume you did, following e.g. [Hoffmann et al. (2022), Section D.2 in appendix](https://proceedings.neurips.cc/paper_files/paper/2022/file/c1e2faff6f588870935f114ebe04a3e5-Paper-Conference.pdf)), choose the optimiser, define the logic of choosing the best fit, similar to [Wortsman et al. (2024)](https://arxiv.org/abs/2405.20204). It is clearly a useful thing for the research community, allowing extended, independent analysis, exemplified by [Besiroglu et al. (2024)'s Chinchilla replication scaling](https://arxiv.org/abs/2404.10102). This is especially important for the ML community going forward, as there are more and more scaling laws works, and comparing them directly will get more and more difficult without the actual data and fitting procedures used.

### 2. Please make the comparison against existing scaling laws fair.

It is my impression that this paper uses comparisons between [Ludziejewski et al. (2024)](https://proceedings.mlr.press/v235/ludziejewski24a.html) and [Abnar et al. (2025)](https://arxiv.org/abs/2501.12370) to convince the reader that functional form proposed by these respective works is inferior to one proposed in this work. I think that's misleading. Seeing as the full training and validation sets in this work have significant variations of $S$ and $N_a / N$, limiting them to ranges similar to the works you compared to would be a more direct comparison between the functional forms. Right now, the experiment only implies that the previous scaling laws do not work well for high $S$ and/or $N_a / N$, which is not surprising, nor does it shed new light on the existing work - they were not designed for these conditions. This can be partially achieved by limiting the train and validation datasets to regions to which the respective scaling laws were designed for - but an issue might come up regardless, if there are not enough datapoints after such filtering.

---

## References

- Kaplan, J., et al. (2020). [Scaling Laws for Neural Language Models](https://arxiv.org/abs/2001.08361).

- Hoffmann, J., et al. (2022). [Training compute-optimal large language models](https://proceedings.neurips.cc/paper_files/paper/2022/file/c1e2faff6f588870935f114ebe04a3e5-Paper-Conference.pdf). NeurIPS 2022.

- Ludziejewski, J., et al. (2024). [Scaling Laws for Fine-Grained Mixture of Experts](https://proceedings.mlr.press/v235/ludziejewski24a.html). ICML 2024.

- Wortsman, M., et al. (2024). [Small-scale proxies for large-scale Transformer training instabilities](https://arxiv.org/abs/2405.20204).

- Porian, T., et al. (2024). [Resolving Discrepancies in Compute-Optimal Scaling of Language Models](https://proceedings.neurips.cc/paper_files/paper/2024/file/b6341525cd84f3be0ef203e4d7cd8556-Paper-Conference.pdf). NeurIPS 2024.

- Besiroglu, T., et al. (2024). [Chinchilla Scaling: A replication attempt](https://arxiv.org/abs/2404.10102).

- Abnar, S., et al. (2025). [Parameters vs FLOPs: Scaling Laws for Optimal Sparsity for Mixture-of-Experts Language Models](https://arxiv.org/abs/2501.12370). ICML 2025.

- Ludziejewski, J., et al. (2025). [Joint MoE Scaling Laws: Mixture of Experts Can Be Memory Efficient](https://proceedings.mlr.press/v267/ludziejewski25a.html). ICML 2025.

---

> ### Author Response · Authors · 2025-11-21
> **Responses to Reviewer**
>
> Thank you for your valuable suggestions! We hope our detailed explanations will address your concerns and questions. Below, W refers to Weakness, Q refers to Question and A refers to Answer.
>
> **W1 \& Q1: Clarity needed on the fitting procedure**
>
> **A**: We sincerely appreciate your suggestions.
>
> Your comment, "This is especially important for the ML community going forward, as there are more and more scaling laws works, and comparing them directly will get more and more difficult without the actual data and fitting procedures used" is highly relevant. **Therefore, we have published our fitting data and validation set data (validation loss; $N$, $N_a$, $D$, $G$, $S$) in the supplementary materials, along with the Python script used for the fitting process to facilitate reproducibility.** Additionally, we plan to release the complete training loss for the Warmup+Stable phases, as well as the Decay phase training loss and validation loss, to contribute to the scaling law open-source community.
>
> - Specifically, we first extracted data points from the decay phase (D, validation loss) from the training log files and applied a sliding window of size 10 for smoothing, selecting the minimum validation loss from the last 5 steps. It is important to note that all factors and validation losses are converted into dimensionless values. Based on this processed data, we predefined the scaling law formula for both baseline methods and our model. We used the \texttt{scipy.optimize.curve\_fit} tool (a core tool for nonlinear curve fitting, which applies nonlinear least squares combined with iterative optimization to find the optimal parameters that best fit the observed data) for least-squares fitting. We set the initial values and bounds of the parameters based on existing work and empirical settings. Since the choice of initial parameter values has a significant impact on the final fitting results, we performed a grid search to ensure that the fitting results were reasonable, with the principle that the fitted parameters should not approach the predefined bounds. During the grid search, we found that regardless of the initial settings, the optimized hyperparameters were proximal in all reasonable fittings.
>
> - The concern regarding "whether the validation set was used for final model selection" did not arise in our fitting process. As described in our paper, we did not select the optimal fitting result based on the validation set's validation loss. The validation set was only used for verifying the extrapolation of our joint scaling law of MoE after scaling up the model.
>
> - The statement "baseline MoE scaling laws were first re-fitted with hyperparameters using the same data points, followed by prediction" means that we did not directly use the formulas from Krajewski [1] and Abnar S [2] to predict the RMSE for the validation set. This would be unfair due to differences in model settings and data that could affect the loss. In fact, to ensure a fair comparison, we used the fitted data from our experiments to perform hyperparameter fitting for both the baseline joint scaling law and ours, and then predicted the validation set data based on the fitted results. This ensured that the differences between methods were solely due to the joint scaling law formula itself. The results show that our joint scaling law outperforms both Krajewski [1] and Abnar S [2] on the validation set (RMSE: 0.0060 vs 0.0193 and 0.0170). Directly using the formulas and hyperparameters from Krajewski [1] and Abnar S [2] to predict the validation set yielded poor results, with RMSEs of 994.5655 and 0.2371, respectively. We are also unable to perform fittings using their original data, as the relevant data have not been publicly disclosed.
>
> (continued on next comment)

---

> ### Author Response · Authors · 2025-11-21
> **Responses to Reviewer**
>
> (Continued from previous comment)
>
> We did not select the optimal fitting result based on the validation set's validation loss. The validation set was only used for verifying the extrapolation of our joint scaling law of MoE. Our scaling law considers a more comprehensive set of factors, which is one of our advantages. Objectively, we have considered five factors that impact the MoE. Thus, the adoption of more hyperparameters compared with the previous MoE scaling laws is scientifically justified. To prevent the risk of overfitting due to too many hyperparameters, we extensively validated our joint scaling law with larger values of $N$, $N_a$, $D$, and different values for the structural factors $G$ and $S$, proving that it is not over-parameterized. Additionally, when designing the scaling law, we applied Occam's razor by not using overly complex formulas for fitting. For example, in Subsection 4.2, for the exponential hyperparameters of $N$ and $N_a$, we found that their fitting results did not differ much, so we represented them with a single hyperparameter. In Subsection 4.3, regarding the form of the check function for $G$, the exponent of $G$ tends to 1, so we did not assign a hyperparameter for the exponent of $G$.
>
> In addition to providing RMSE and visual comparisons, we also provide derivations based on the joint scaling law and its application to industrial open-source models in Appendices D, E, and F. Furthermore, in Appendix G, we analyze the optimal loss for different total computation costs $C$. We also include comparisons of the fitted hyperparameters in Table 6 and 7 of Appendix I. Besiroglu et al.’s analysis methods have been considered [3], including loss error analysis, factors and total computation cost $C$ analysis, and hyperparameter fitting result comparisons.
>
> **W2: Clarification needed on validation set characteristics**
>
> **A**: In fact, the validation set relative to the training data points involves both interpolation and extrapolation. When designing the experiments, we defined several classic MoE models of different sizes, as shown in Table 1. For different factors, we performed controlled variable scaling experiments on smaller-sized models, and then validated on larger sizes. To better understand the distribution of factors in the fitted data and validation set, we have included distribution plots in Appendix L, Figure 18. Specifically, for fitting data points, $G \in (1, 20]$, $S \in [0.0, 0.8]$, $N_a \in [30\text{M}, 2.2\text{B}]$, $N \in [133\text{M}, 3.4\text{B}]$ and $D \in [10\text{B}, 50\text{B}]$; for validation data points, $G \in (1, 40]$, $S \in [0.0, 0.7]$, $N_a \in [453\text{M}, 6.6\text{B}]$, $N \in [2.4\text{B}, 30\text{B}]$ and $D \in [10\text{B}, 100\text{B}]$.
>
> **Additionally, we supplemented experiments with $N30BA3B$ (data size $D = 50B$) and $N907MA181M$ (data size $D = 10B, 20B, 50B$) with $G = 40$, to verify the extrapolation of our joint scaling law of MoE for larger models and $G$.** Specifically, for the N30B validation data, ours vs Krajewski [1] vs Abnar S [2]: 0.0007 vs 0.0073 vs 0.0200; for $G = 40$ validation data, ours vs Krajewski [1] vs Abnar S [2]: 0.0085 vs 0.0647 vs 0.0702.

---

> ### Author Response · Authors · 2025-11-21
> **Responses to Reviewer**
>
> **W3 \& Q2: Suggestion for distributional comparison with prior work**
>
> **A**: Thank you for your suggestions for distributional comparison with prior work. Firstly, our experimental design, including model results, data size, and the range of factor parameters, aligns with actual parameter designs. In fact, our comparison experiments are fair. We used the fitted data from our experiments to perform hyperparameter fitting for both the joint scaling law of the baselines and ours, and then predicted the validation set data based on the fitted results. This ensures that the differences between methods are solely due to the joint scaling law formula itself. Our joint scaling law is a comprehensive scaling law, and considering more factors and appropriately fusing them into a joint scaling law is one of our advantages. In fact, these factors significantly affect the performance of MoE models and need to be considered. A good MoE scaling law must address the loss prediction issue across a wide range of factor settings.
>
> Additionally, in Krajewski's experiments [1], the values of G are [1, 2, 4, 8, 16], while in AbnarS's experiment [2], the range of Sparsity is from 0\% to 100\%, achieved by adjusting the activated experts. The range of G both varies widely. In our experiment, the range of G is from 1 to 20, so the distributions of G across the three studies are quite similar. As for S, neither Krajewski [1] nor AbnarS [2] addressed it. However, considering that the data with $S = 0$ is relatively sparse, and taking into account that the Shared Expert setup is likely to be widely adopted, we thus selected $S \leq 0.2$ as the threshold to filter the fitting data and validation set data. Specifically, ours vs Krajewski [1] vs Abnar S [2]: 0.0075 vs 0.0331 vs 0.0147. The experimental results show that our joint scaling law of MoE significantly outperforms other baselines even under their typical parameter distributions.
>
> **W4: Minor concern regarding experimental controls**
>
> **A**: We greatly appreciate your suggestions on the experimental configurations of batch size and learning rate. The primary focus of this paper is on scaling factors related to model size, architecture, and data size. While the learning rate is not within the scope of our current study, we acknowledge its importance. On one hand, we fixed the batch size and learning rate to reasonable values to minimize bias. Additionally, we avoided very small-scale experiments when designing the experiments. On the other hand, considering both batch size and learning rate would double the experimental cost, which is beyond our current budget. Your suggestion is valid: a comprehensive scaling law should ideally include variables such as data size, total model size, activated model size, number of active experts, ratio of shared experts, learning rate, batch size, and others. In future work, we plan to use existing scaling laws for batch size and learning rate to guide our selection of these parameters, as Step Law [4] suggests that learning rate is related to $N$ and $D$, while batch size is related to $D$.
>
>
> **References**
>
> [1] Krajewski J, Ludziejewski J, Adamczewski K, et al. Scaling laws for fine-grained mixture of experts[J]. arXiv preprint arXiv:2402.07871, 2024.
>
> [2] Abnar S, Shah H, Busbridge D, et al. Parameters vs flops: Scaling laws for optimal sparsity for mixture-of-experts language models[J]. arXiv preprint arXiv:2501.12370, 2025.
>
> [3] Besiroglu T, Erdil E, Barnett M, et al. Chinchilla scaling: A replication attempt[J]. arXiv preprint arXiv:2404.10102, 2024.
>
> [4] Li H, Zheng W, Hu J, et al. Predictable Scale: Part I--Optimal Hyperparameter Scaling Law in Large Language Model Pretraining[J]. arXiv e-prints, 2025: arXiv: 2503.04715.

---

### Official Review · Reviewer_WRLL · 2025-10-31

**Soundness:** 2
**Presentation:** 3
**Contribution:** 3
**Rating:** 6
**Confidence:** 3

**Summary:**

The work proposes a scaling law for Mixture of Experts decoder-only Transformers, which unifies the following factors:
- number of training tokens ($D$),
- total number of model parameters ($N$),
- number of active parameters per token ($N_a$),
- number of active experts per token ($G$),
- fraction of shared experts among all active experts ($S$).
The paper walks through the derivation of the scaling law, starting with the Chinchilla-like Scaling Law (dependent on $N$, $D$) and expanding it by considering the impact of: $N_a$, $G$, and finally, $S$. The quality of the fit is compared to two earlier works ([3, 4]). Based on the final law, some key implications are derived:
- The optimal number of activated experts $G$ (which is constant)
- The optimal ratio of shared experts $S$ (which is constant)
- The optimal sparsity (which increases as models grow).

**Strengths:**

- Large number and variety of performed experiments
- Principled, step-by-step description of the process of deriving the scaling law, including fitting the scaling laws for intermediate formulas
- Clear presentation of the key implications (section 5)

**Weaknesses:**

- The paper does not specify whether embedding or non-embedding active / total parameters are used
- There is no bootstrapping involved, i.e. no measure of uncertainty for the fitted coefficients in the Table 2
- All experiments use the same learning rate, which is not uncommon in scaling laws research, however, other approaches have been proposed [2, 3]
- Lack of evaluation on downstream tasks
- Omission of some technical details (RMSNorm or LayerNorm, vocab size, positional encodings used, etc.), leading to reduced reproducibility. Supplying the code would contribute to improving this aspect.
- Limited information on the specifics of the fitting procedure (what algorithm was used to optimize the coefficients, what was the grid of initial values in fitting, etc.)
- It is not clear how many experts per layer there are in the models: line 146 says "up to 256 experts", however table 1 claims 32 experts for all models.
- The notion of "Efficiency awareness" is not explained clearly (see questions)
- If I understand correctly, the presented scaling law does not consider scaling of $E$ (expansion rate as per [3]). This is a clear limitation which constrains the applicability of the results (see [2]) and __needs__ to be mentioned in the limitations section (and perhaps in the table 5 as well).

Other minor remarks:
- C.1 title should read "Numerical" not "Numberical"
- Line 492 incorrectly claims the appendix F contains information about the implementation details

**Questions:**

- OLMoE (Muennighoff et al., https://arxiv.org/pdf/2409.02060v1) fail to find convincing evidence for employing shared experts. Do the authors have an idea how to reconcile that result with the results in this work?
- Can the authors clarify the usage of embedding / non-embedding parameters?
- It seems a standard practice (Scaling Laws for Neural Language Models, Kaplan et al., https://arxiv.org/pdf/2001.08361; Joint MoE Scaling Laws: Mixture of Experts Can Be Memory Efficient, Ludziejewski et al., https://arxiv.org/pdf/2502.05172v2) is to keep constant depth-to-width (dm / l) ratio for all models, however here the ratio changes (Table 1). Is there a principled reason for that?
- What are the "associated costs" mentioned in line 418?
- What is the applicability of the results considering the lack of scaling w.r.t. $E$ (expansion rate, see [3])?

References:
1. Scaling Laws for Neural Language Models, Kaplan et al., https://arxiv.org/pdf/2001.08361
2. Joint MoE Scaling Laws: Mixture of Experts Can Be Memory Efficient, Ludziejewski et al., https://arxiv.org/pdf/2502.05172v2
3. Scaling Laws for Fine-Grained Mixture of Experts, Krajewski et al., https://arxiv.org/abs/2402.07871
4. Parameters vs FLOPs: Scaling Laws for Optimal Sparsity for Mixture-of-Experts Language Models, Abnar et al., https://arxiv.org/abs/2501.12370

---

> ### Author Response · Authors · 2025-11-21
> **Responses to Reviewer**
>
> Thank you for your valuable feedback and insightful guidance in improving our work. We have provided the following responses to address your comments and questions. "W" refers to Weakness, "Q" refers to Question and "A" refers to Answer. Similar questions have been grouped together. All new updates on paper are colored in blue.
>
> **W1 & Q2: The paper does not specify whether embedding or non-embedding active / total parameters are used. (Q2: Can the authors clarify the usage of embedding / non-embedding parameters?)**
>
> **A**: Thanks for pointing out that. In all of our experiments, the parameters considered do not include those from the embedding layer. We have now included the necessary clarifications in Appendix B.
>
> **W2: There is no bootstrapping involved, i.e. no measure of uncertainty for the fitted coefficients in the Table 2.**
>
> **A**: Our fitting process is deterministic, and as such, we do not incorporate uncertainty estimation for the coefficients. We use the least squares method for fitting, and further details on the fitting data and procedures are available in the supplementary materials. Additionally, similar studies in the field [1, 2, 3, 5, 6, 7] generally do not report uncertainty measures for the fitted coefficients.
>
> **W3: All experiments use the same learning rate, which is not uncommon in scaling laws research, however, other approaches have been proposed.**
>
> **A**: Thanks for your valuable comment. The primary focus of this paper is on scaling factors specific to MoE models, such as $G$, $N_a$, $S$, and their impact on the loss function. Most works on MoE scaling laws have also overlooked the learning rate [1, 2, 3, 4, 5, 6].
> While the learning rate is beyond the scope of our current study, we acknowledge its importance and agree that it should be considered in a more comprehensive scaling law, which we plan to explore in future work. However, if we consider all factors, we face challenges in terms of fitting difficulty, experimental complexity and cost, as well as the generalizability of the resulting formulas.
>
> **W4: Lack of evaluation on downstream tasks.**
>
> **A**: Our study primarily focuses on loss prediction, a strong indicator of foundation model's performance. It is a key method for evaluating both model quality and generalizability. Additionally, loss during model training is not only more stable but also easier to extrapolate. Loss prediction has also been adopted as a critical metric in several related studies [1, 2, 3, 4, 5, 6, 7]. And there is limited work on scaling laws that quantifies the relationship between downstream task metrics and pretraining settings.
> We also acknowledge that evaluation on downstream tasks is important. However, this approach has its challenges. Downstream task metrics can be influenced by a variety of factors, including task type, data for SFT and RL, training algorithms, and scheduling strategies. And when loss values are similar, it can be difficult to distinguish downstream tasks performance. Additionally, the performance on downstream tasks tends to be more unstable, making it less reliable for predictions. We plan to explore downstream task evaluations in future research.
>
> **W5: Omission of some technical details (RMSNorm or LayerNorm, vocab size, positional encodings used, etc.), leading to reduced reproducibility. Supplying the code would contribute to improving this aspect.**
>
> **A**: We appreciate your valuable feedback. We have provided typical settings of our experiments in Table 1 of Appendix B. We also have provided more details you mentioned in Table 1. What's more, we have made our fitting data and fitting procedure available at the supplementary materials. Additionally, we will release more codes and related configurations in the future to further enhance the reproducibility of our experiments.

---

> ### Author Response · Authors · 2025-11-21
> **Responses to Reviewer**
>
> **W6: Limited information on the specifics of the fitting procedure (what algorithm was used to optimize the coefficients, what was the grid of initial values in fitting, etc.)**
>
> **A**: Specifically, we first extracted the data points from the decay phase of the training log files (i.e., $D$, validation loss) and applied a sliding window of size 10 for smoothing, selecting the minimum loss over the last 5 steps as the validation loss. It is important to note that all factors and validation losses were scaled to dimensionless values. Based on the processed data, we predefined the scaling law formula for both baseline methods and ours. We used \texttt{scipy.optimize.curve\_fit} (a core tool for nonlinear curve fitting using nonlinear least squares, which iteratively optimizes to find the best-fit model parameters) for least-squares fitting. The initial parameter values and bounds were set based on existing work and empirical settings. Since the choice of initial parameter values significantly impacts the final fitting results, we performed a grid search to determine whether the fitting results were reasonable, with the principle that the fitted parameters should not approach the predefined bounds. During the grid search, we found that, regardless of the reasonable initial settings, the optimized hyperparameters obtained through least-squares fitting were consistent across different initial values. **We have provided our fitting data and validation loss ($N$, $N_a$, $D$, $G$, $S$) in the supplementary materials, along with the Python script used for the fitting process.**
>
> **W7: It is not clear how many experts per layer there are in the models: line 146 says "up to 256 experts", however table 1 claims 32 experts for all models.**
>
> **A**: The information in Table 1 of Appendix B reflects typical configurations. When conducting controlled variable experiments on factors $G$, $N$, and $N_a$ in our study, the number of experts changes accordingly. The total number of experts in our experiment ranges from [32, 64, 128, 256], ensuring the generalizability of the scaling law. For specific details on the variation of different factors, please refer to Appendix K.
>
> **W8: The notion of "Efficiency awareness" is not explained clearly (see questions)**
>
> **A**:  Thanks for your valuable comment. To clarify "efficiency awareness," consider the activated model size ($N_a$). As $N_a$ increases and $N_a/N$ grows, the loss first decreases, then increases, indicating an optimal value. However, at this optimal point, $N_a$ is large, leading to higher training and inference costs. On the left side of the optimal point, costs are more manageable while the loss remains sufficiently low. To address this, we define a threshold ($\Delta Loss = 0.001$ or $\Delta Loss = 0.005$) where the difference between the current and optimal loss is within an acceptable range, allowing us to recommend a feasible $N_a$ value. More detailed analysis can be found in Section 5.3 and Appendix F.
>
> **W9 & Q5: If I understand correctly, the presented scaling law does not consider scaling of $E$ (expansion rate as per [3]). This is a clear limitation which constrains the applicability of the results (see [2]) and needs to be mentioned in the limitations section (and perhaps in the table 5 as well). (What is the applicability of the results considering the lack of scaling w.r.t.  (expansion rate, see [3])?)**
>
> **A**: Thanks for your comment. In fact, once $N$, $N_a$, $S$, and $G$ are determined, $E$ is also fixed. Therefore, there is no need to introduce $E$ separately, as it can be directly determined from the existing factors. Therefore, there is no issue with the applicability of our approach.
>
> **W10: C.1 title should read "Numerical" not "Numberical"**
>
> **A**: Thanks for pointing out it. We have corrected it in the paper.
>
> **W11: Line 492 incorrectly claims the appendix F contains information about the implementation details**
>
> **A**: Thanks for pointing out it. The implementation details are primarily provided in Table 1, which presents some typical settings and hyperparameter configurations for our experiments. Additional experimental configurations and details can be found in Table 8 in Appendix L. **Furthermore, we have made our fitting data and fitting procedure available at the supplementary materials.** Additionally, we will release more codes and related configurations in the future to further enhance the reproducibility of our experiments.

---

> ### Author Response · Authors · 2025-11-21
> **Responses to Reviewer**
>
> **Q1: OLMoE (Muennighoff et al., [https://arxiv.org/pdf/2409.02060v1](https://arxiv.org/pdf/2409.02060v1)) fail to find convincing evidence for employing shared experts. Do the authors have an idea how to reconcile that result with the results in this work?**
>
> **A**: Thanks for your question. We make following explanations in response.
> (a) In our experiments, the range of $S$ varies from 0 to 0.7 across different model sizes, and our results consistently show that shared experts are indeed beneficial.
> (b) Many open-source models like DeepSeek-MoE [8], DeepSeek-V3 [9], GLM-4.5 [10], and Kimi-K2 [11], have also use this mechanism.
> (c) Shared experts capture common knowledge across diverse contexts. By compressing shared knowledge, redundancy among the other routed experts is effectively reduced, resulting in enhancement to the model’s performance.
>
> **Q3: It seems a standard practice (Scaling Laws for Neural Language Models, Kaplan et al., [https://arxiv.org/pdf/2001.08361](https://arxiv.org/pdf/2001.08361); Joint MoE Scaling Laws: Mixture of Experts Can Be Memory Efficient, Ludziejewski et al., [https://arxiv.org/pdf/2502.05172v2](https://arxiv.org/pdf/2502.05172v2)) is to keep constant depth-to-width (dm / l) ratio for all models, however here the ratio changes (Table 1). Is there a principled reason for that?**
>
> **A**: We appreciate the reviewer for raising this point and make following explanations.
> (a) We have calculated the width / depth of the models involved in our experiments, and these values fall within a reasonable range, as shown in Table A below.
> (b) Additionally, we examine the width / depth ratios in open-source model families such as the Qwen series, as shown in Table B.
> (c) In Joint MoE Scaling Laws: Mixture of Experts Can Be Memory Efficient, the width / depth = 64. As can be seen, the width/depth ratios in our experiments are within a reasonable range. In our study, we focused on variables such as $N$, $N_a$, $G$, $S$, and $D$, and conducted experiments under reasonable settings to investigate the scaling laws.
> (d) Furthermore, other related works [3, 4, 5, 6, 7] have not emphasized this factor or not controlled this factor.
>
> | Total model size     | 247M  | 496M | 907M  | 2.40B | 3.96B |
> | -------------------- | ----- | ---- | ----- | ----- | ----- |
> | Activated model size | 48M   | 99M  | 181M  | 476M  | 793M  |
> | depth                | 12    | 12   | 12    | 20    | 24    |
> | width                | 512   | 768  | 1024  | 1280  | 1536  |
> | width / depth        | 42.67 | 64   | 85.33 | 64    | 64    |
> Table A: width / depth of our experimental settings.
>
> | Model Name    | Qwen3-0.6B | Qwen3-1.7B | Qwen3-4B | Qwen3-8B | Qwen3-14B | Qwen3-32B |
> | ------------- | ---------- | ---------- | -------- | -------- | --------- | --------- |
> | depth         | 28         | 28         | 36       | 36       | 40        | 64        |
> | width         | 1024       | 2048       | 2560     | 4096     | 5120      | 5120      |
> | width / depth | 36.57      | 73.14      | 71.11    | 113.78   | 128       | 80        |
> Table B: width / depth of Qwen3 series models.
>
> **Q4: What are the "associated costs" mentioned in line 418?**
>
> **A**: Thank you for your question. To clarify, we are referring to the associated costs in terms of the value of $N_a$. As $N_a$ increases, both the training cost of the model and the cost of deploying and performing inference with the model also rise.
>
> We hope that our response has satisfactorily addressed your concerns. We welcome any further questions or suggestions and appreciate your feedback in helping us improve.
>
> **References**:
>
> [1] Scaling Laws for Neural Language Models, Kaplan et al., [https://arxiv.org/pdf/2001.08361](https://arxiv.org/pdf/2001.08361)
>
> [2] Joint MoE Scaling Laws: Mixture of Experts Can Be Memory Efficient, Ludziejewski et al., [https://arxiv.org/pdf/2502.05172v2](https://arxiv.org/pdf/2502.05172v2)
>
> [3] Scaling Laws for Fine-Grained Mixture of Experts, Krajewski et al., [https://arxiv.org/abs/2402.07871](https://arxiv.org/abs/2402.07871)
>
> [4] Parameters vs FLOPs: Scaling Laws for Optimal Sparsity for Mixture-of-Experts Language Models, Abnar et al., [https://arxiv.org/abs/2501.12370](https://arxiv.org/abs/2501.12370)
>
> [5]Predictable scale: Part i–optimal hyperparameter scaling law in large language model pretraining., https://arxiv.org/abs/2503.04715
>
> [6]Scaling laws for floating point quantization training., https://arxiv.org/abs/2501.02423
>
> [7]Unified scaling laws for routed language models., https://arxiv.org/abs/2202.01169
>
> [8] DeepSeekMoE: Towards Ultimate Expert Specialization in Mixture-of-Experts Language Models., https://arxiv.org/abs/2401.06066
>
> [9] DeepSeek-V3 Technical Report., https://arxiv.org/abs/2412.19437v1
>
> [10] GLM-4.5: Agentic, Reasoning, and Coding (ARC) Foundation Models., https://arxiv.org/abs/2508.06471
>
> [11] Kimi K2: Open Agentic Intelligence., https://arxiv.org/abs/2507.20534

---

### Official Review · Reviewer_NbT9 · 2025-11-02

**Soundness:** 3
**Presentation:** 3
**Contribution:** 3
**Rating:** 6
**Confidence:** 4

**Summary:**

This paper proposes a scaling law for MoE language models, extending scaling relations to include five interacting factors: total model size N, data size D, activated model size Na, number of active experts G, and shared-expert ratio S. The authors empirically derive a joint loss function that accurately predicts performance across a wide range of MoE configurations and yields optima such as  Gopt ⁣≈ ⁣7, Sopt ⁣≈ ⁣0.3, and decreasing optimal activation ratio.

**Strengths:**

The paper presents one of the most extensive empirical studies of MoE scaling , systematically analyzing five interrelated factors—
D,N,Na,G,S

The authors methodically vary one factor at a time, isolate marginal effects, and then assemble a joint scaling law.

The implications are interesting, leveraging the quadratic equation model to obtain optimal G and S and that the N_a/N decreases with total model size. The derived optima correspond well to empirical settings in industrial MoEs

The model fit is very good.

**Weaknesses:**

The main findings are about S and G which have relatively little impact for the loss. Due to small impact on S, it is questionable if they should be included in the formula, making it seem a little over-engineered.

The paper lacks the validation of the scaling law on a very large scale (e.g. much larger model that was not used to get the scaling law)

No analysis of variance across random seeds. Given the many fitted coefficients, it’s unclear whether the final joint law is over-parameterized or robust to noise.

A little confusing notation, G originally described granularity

**Questions:**

You include the parameters of the number of acivated experts but not the number of experts and no granularity. How did you make choices of parameters to appear in your scaling law?

How do the fitted exponents parameters (say α,β) compare with those in other works, Krajewski, Ludziejewski, Chinchilla?

---

> ### Author Response · Authors · 2025-11-21
> **Responses to Reviewer**
>
> Thank you for your valuable suggestions! We hope that our detailed explanations will address your concerns and questions. Below, W refers to Weakness, Q refers to Question and A refers to Answer.
>
> **W1: The main findings are about S and G which have relatively little impact for the loss. Due to small impact on S, it is questionable if they should be included in the formula, making it seem a little over-engineered.**
>
> **A**: We sincerely appreciate your suggestions. Indeed, $S$ and $G$ are structural factors specific to MoE models. Compared to $N$, $N_a$, and $D$, their impact on the loss is smaller, but $G$ and $S$ still significantly affect the loss. Specifically, Figure 3 shows that different values of $G$ result in percentile-level impacts on the loss, and Figure 4 demonstrates that varying values of $S$ lead to millesimal-level changes. Therefore, it is necessary to study the impact of structural factors ($G$ and $S$) on the scaling law. The influence of structural factors on MoE models has been confirmed in related studies[1,2]. Indeed, considering the relatively small impact of $S$ on the loss, we provide two sets of joint scaling laws of MoE in Sections 4.4 and 4.5 of the paper: one with $S$ and one without, allowing other researchers in the community to select accordingly.
>
> Furthermore, the statement that "$S$ has a small impact" is somewhat limited. In Appendix C.5, Figure 14, we find that the impact of $S$ on the loss becomes more pronounced as the model size scales (from $N_a48M$-$N247M$ to $N_a793M$-$N3964M$), with the effect of $S$ ranging from 0.0014 to 0.0181. This clearly demonstrates the necessity of considering and studying the scaling of $S$, especially for current industrial LLMs with parameter sizes of 1T, 2T, and beyond, such as Kimi K2 [3], ERNIE-5.0, and Grok 4. Based on our fitting results, whether comparing experiments in Figure 5 or ablation studies in Appendix C, our comprehensive scaling law of MoE provides more accurate fits. In future work, we plan to investigate the marginal effects of $S$ on even larger models to further validate the correctness of our joint scaling law of MoE.
>
> **W2: The paper lacks the validation of the scaling law on a very large scale (e.g. much larger model that was not used to get the scaling law)**
>
> **A**: To validate the scaling law on a very large scale, **we have added experiments with N30BA3B, where the data size $D$ is 50B, to test the extrapolation and robustness of our joint scaling law of MoE.** The results of these experiments have been included in Figure 5 and Table 8 of the paper, with the relevant results marked by a black border. The results show that our scaling law remains effective for the N30BA3B size (ours vs Krajewski [4] vs Abnar S [5]: 0.0007 vs 0.0073 vs 0.0200). Moreover, the model sizes of the validation set data points are larger than those used for fitting, which partially verifies the extrapolation of our scaling law to larger models. We hope these additional experiments address your concerns.

---

> ### Author Response · Authors · 2025-11-21
> **Responses to Reviewer**
>
> **W3: No analysis of variance across random seeds. Given the many fitted coefficients, it’s unclear whether the final joint law is over-parameterized or robust to noise.**
>
> **A**: We will explain the concerns regarding "random seed" and "over-parameterization" separately.
>
> - Due to the high experimental cost associated with scaling law experiments, we fixed the random seed to 1234 for most of these experiments. In selecting the validation loss, we applied a sliding window method to smooth the data, which provides some degree of robustness against noise. **In fact, our fitting data includes multiple experiments with the same configuration but different random seeds. To aid researchers in understanding the results and contribute to the scaling law open-source community, we have included the fitting and validation data (validation loss; N, $N_a$, D, G, S) in the supplementary materials, along with the Python script used for the fitting process.** The experiments with different seeds are clearly labeled. For example, under the experimental setting $N247M-N_a48M-D50B-G10-S0.2$, the validation losses for seeds 1234, 1122, and 3344 were 2.9503, 2.9504, and 2.9509, respectively. We computed the average standard deviation of all experiments with different seeds, which was 0.000849. The results indicate that the random seed has minimal impact on the validation loss. We hope that this additional explanation addresses your concerns.
>
>
> - Regarding the concern about over-parameterization, we have extensively validated our joint scaling law with larger values of $N$, $N_a$, and $D$, as well as with different values for the structural factors $G$ and $S$, proving that the model is not over-parameterized. Additionally, when designing the scaling law, we paid special attention to applying Occam’s razor by not using overly complex formulas for fitting. For example, in Subsection 4.2, for the exponential hyperparameters of $N$ and $N_a$, we found that their fitting results did not differ much and thus represented them with a single hyperparameter. In Subsection 4.3, regarding the form of the check function for $G$, the exponent of $G$ tends to 1, so we did not assign a hyperparameter for the exponent of $G$.
>
> **W4: A little confusing notation, G originally described granularity**
>
> **A**: In our paper, $G$ is defined as the number of activated shared and routed experts. As shown in Appendix K, Figure 17(a), when the number of activated shared and routed experts scales, the expert dimensions are divided into finer granularity to maintain other factors unchanged, which aligns with the concept of "granularity." Therefore, we use $G$ as an abbreviation.
>
> **Q1: You include the parameters of the number of acivated experts but not the number of experts and no granularity. How did you make choices of parameters to appear in your scaling law?**
>
> **A**: It is a significant issue. When selecting factors that affect MoE, we consider both parameters/data volume and structural factors, including $N$, $N_a$, $D$, and $G$, $S$. $G$ essentially reflects the concept of granularity. In our controlled experiments, as $G$ increases, the expert dimensions become more finely divided. In the experiments controlling $N$, as $N$ increases, the number of routed experts gradually increases. Thus, our scaling law can reflect both the number of experts and granularity, as you mentioned.
>
> **Q2: How do the fitted exponents parameters (say α,β) compare with those in other works, Krajewski, Ludziejewski, Chinchilla?**
>
> **A**: We greatly appreciate your question. Different pretraining datasets lead to different fitted hyperparameters, so directly comparing the numerical values of hyperparameters across different data sources is not meaningful. As described in Appendix I.3, to effectively compare the fitted hyperparameters, we re-fit the scaling law of MoE by Krajewski [4] and Abnar S [5] using our experimental data points (validation loss vs $N$, $Na$, $D$, $G$, $S$) and then perform the comparison. To better compare the fitted hyperparameter values, we have included the fitting results in Table 6 and 7 of Appendices I.1 and I.2. These results show the values of $\alpha$, $\beta$, and $\epsilon$ as follows: (1) Ours (0.2383, 0.4694, 1.8182); (2) Krajewski (0.2380, 0.3695, 1.771); (3) Abnar S (0.1948, 0.4145, 1.6351). Additionally, we compared the hyperparameters from another work, "Scaling Laws for Floating-Point Quantization Training" [6], which used the Dolma 1.7 dataset. Specifically, their fitted values for $\alpha$, $\beta$, and $\epsilon$ were (0.2368, 0.5162, 1.9061). Overall, hyperparameter fittings have slight variations due to different scaling law formulas but are overall relatively close.

---

> ### Author Response · Authors · 2025-11-21
> **Responses to Reviewer**
>
> **References**
>
> [1] Tian C, Chen K, Liu J, et al. Towards greater leverage: Scaling laws for efficient mixture-of-experts language models[J]. arXiv preprint arXiv:2507.17702, 2025.
>
> [2] Liu A, Feng B, Xue B, et al. Deepseek-v3 technical report[J]. arXiv preprint arXiv:2412.19437, 2024.
>
> [3] Team K, Bai Y, Bao Y, et al. Kimi k2: Open agentic intelligence[J]. arXiv preprint arXiv:2507.20534, 2025.
>
> [4] Krajewski J, Ludziejewski J, Adamczewski K, et al. Scaling laws for fine-grained mixture of experts[J]. arXiv preprint arXiv:2402.07871, 2024.
>
> [5] Abnar S, Shah H, Busbridge D, et al. Parameters vs flops: Scaling laws for optimal sparsity for mixture-of-experts language models[J]. arXiv preprint arXiv:2501.12370, 2025.
>
> [6] Sun X, Li S, Xie R, et al. Scaling laws for floating point quantization training[J]. arXiv preprint arXiv:2501.02423, 2025.

---

### Author Response · Authors · 2025-12-03
**Summary of Rebuttal**

Dear Area Chair

We sincerely appreciate the time and effort you have dedicated to reviewing our paper. To help you quickly and thoroughly grasp the key points of our rebuttal, we provide a concise summary below outlining how we have addressed the reviewers' major concerns.

### Key Contribtuions

Existing scaling laws for dense models are inapplicable to MoE models, which stems from three critical challenges: the multiplicity of influencing factors, their intricate coupling relationships and the non-monotonic nature of their performance impacts. In this work, we perform a systematic decomposition of MoE settings, identifying five key factors that influence model performance from both size and structural perspectives (data size ($D$), total model size ($N$), activated model size ($N_a$), number of active experts ($G$) and the ratio of shared experts ($S$)). Specifically, **we design $450$ controlled experiments** to characterize their marginal effects, ultimately constructing a comprehensive and precise joint MoE scaling law that considers all essential factors. Furthermore, we derive the theoretically optimal and practically efficiency-aware optimal configurations for $G$, $S$ and $N_a / N$ with detailed analyses. Our results demonstrate that the optimal settings for $G$ and $S$ are independent of both the model architecture and data size. With the scaling of $N$, the optimal activation parameter ratio of $N_a/N$ becomes sparser. Our proposed MoE scaling law could function as an accurate and insightful guidance to facilitate future MoE model design and training.

---

> ### Author Response · Authors · 2025-12-03
>
> ### Summary of Rebuttal
>
> The following table provides a summary of our rebuttal. For detailed responses, refer to the specific Reply to Reviewers.
>
> |Reviewer|Strengths|Main Weaknesses & Questions|Initial Rate|Our Reply|
> |-|-|-|-|-|
> |**NbT9**| **S1**: Most extensive empirical study of MoE scaling **S2**: Methodical approach.**S3**: Interesting implications of the optimal G, S and Na/N; derived optima align with industrial MoEs.**S4**: Very good model fit| 🔥**W1 & Q1:** Selection of MoE factors, in particular $S$ and $G$.🔥**W2:** Lacks validation on very large scale models.🔥**W3:** No variance analysis across random seeds; unclear if model is over-parameterized or robust to noise| **6**| ✅**W1 & Q1:** S and G are MoE-specific structural factors with significant impacts (S is measured by percentile, G by millesimal). The impact of S increases with scale (rising from 0.0014 to 0.0181). Selected factors include parameters (N, Na, D) and structural factors (G, S). G reflects granularity through the division of expert dimensions, while S reflects the number of experts. In our paper, two versions of the scaling law are provided (with and without S).✅**W2:** **N30BA3B experiments (trained on 50B data) were supplemented for large-scale validation.** Results demonstrate effectiveness, with errors of 0.0007, 0.0073, and 0.0200 (ours vs. comparative baselines).✅**W3:** To control computational costs, a fixed random seed (seed=1234) was used in most experiments. **Additionally, supplementary several experiments with different random seeds were conducted.** The results demonstrate minimal variance (average standard deviation = 0.000849).|
> **WRLL** | **S1:** Large number and variety of performed experiments.**S2:** Principled, step-by-step description of deriving the scaling law.**S3:** Clear presentation of key implications| 🔥**W3:** All experiments use same learning rate; other approaches exist.🔥**W9 & Q5:** Doesn't consider expansion rate scaling.🔥**Q1:** Necessity of shared experts.🔥**Q3:** Why not keeping constant depth-to-width ratio (dm/l). |6| ✅**W3:** Focus is on MoE-specific factors (Na, G, S). Most MoE scaling law works overlook learning rate. Acknowledge importance for future work, but considering all factors poses challenges in fitting difficulty, experimental cost, and generalizability.✅**W9 & Q5:** Once Na, N, D, S determined, expansion rate is fixed. No need to introduce $E$ separately - directly determined from existing factors. No applicability issue.✅**Q1:** (a) Our experiments show S ∈ [0, 0.7] - shared experts are beneficial. (b) Many models use this: DeepSeek-MoE, DeepSeek-V3, GLM-4.5, Kimi-K2. (c) Shared experts capture common knowledge, reduce redundancy, enhance performance.✅**Q3:** (a) Our width/depth ratios are reasonable (42.67-85.33). (b) Qwen series also varies (36.57-128). (c) Joint MoE Scaling Laws uses 64; ours are comparable. (d) Related works don't emphasize this factor. Our work mainly focused on Na, N, D, G, S variables. |
> | **f3c7** | **S1:** Broad scope, covering key architectural and scaling choices in MoE LLMs.**S2:** Systematic hypothesis development, from dense to joint scaling laws.**S3:** Extensive experimental design, with 446 controlled experiments.**S4:** Analytical interpretation, deriving optimal Na, G, S, and comparisons with large MoE models.**S5:** Clear reporting, with all experimental configurations provided. | 🔥**W1 & Q1:** Clarity needed on the fitting procedure.🔥**W2:** Clarification needed on validation set characteristics.🔥**W3:** Suggestions for distributed Comparison with prior work.🔥**W4:** Minor concern regarding experimental controls| 6| ✅**W1 & Q1**: **We published fitting data, validation set, and Python scripts in supplementary materials.** Fitting process: (1) Extracted decay phase data using sliding window (size 10); (2) Normalized factors to dimensionless values; (3) Applied scipy.optimize.curve_fit with grid search validation. Validation set was not used for model selection, only for verifying extrapolation. The comparison results are (RMSE: 0.0060 vs 0.0193 vs 0.0170).✅**W2**: Validation includes interpolation and extrapolation. Fitting range and Validation range are detailed in Appendix L. Results show strong extrapolation: N30BA3B (0.0007 vs 0.0073 vs 0.0200); G=40 (0.0085 vs 0.0647 vs 0.0702).✅**W3 & Q2**: Our experimental design reflects realistic configurations. Considering more factors (G, S) is necessary as they significantly impact MoE performance. Distribution comparison shows our ranges (G=[1,20], S∈[0,0.7]) are comparable to prior work. Even filtering to S≤0.2 shows our law outperforms (0.0075 vs 0.0331 vs 0.0147).✅**W4**: Fixed batch size and learning rate at reasonable values to minimize bias and reduce experimental cost. Future work will incorporate learning rate and batch size.|

---

> ### Author Response · Authors · 2025-12-03
> **Summary of Rebuttal**
>
> ### Core Instructions of Rebuttal
> #### Technical and Experimental Detail Clarifications
>
> We sincerely appreciate the reviewers' valuable suggestions. To enhance the clarity and rigor of our paper, we have supplemented additional experimental and technical details. **Specifically, we have included all experimental data points and fitting scripts in the supplementary materials** to further strengthen the credibility and reproducibility of our research. Providing comprehensive data and fitting procedures can advance the MoE Scaling Law research community. Furthermore, this work provides more comprehensive scaling laws, which offer valuable insights for designing more powerful MoE models in the future. We thank the reviewers for their constructive comments.
>
> To better address the reviewers’ comments, provide necessary clarifications, and strengthen the soundness of our paper, we have conducted several supplementary experiments as detailed below:
>
> - Large-scale validation of extrapolation capability. **To validate our MoE joint scaling law at scale, we conducted additional experiments with N30BA3B-D50B to test its extrapolation capability and robustness.** The results confirm the effectiveness of our scaling law for N30BA3B. Importantly, since these validation data points involve larger model sizes than those used for fitting, this partially verifies the law’s extrapolation to larger models. Moreover, our proposed formula demonstrates superior fitting performance on these large-scale model data points compared to baseline approaches (0.0007 vs 0.0073 vs 0.0200). It is worth noting that baseline works rarely extend their experimental model sizes to this scale.
>
> - Extrapolation to larger expert num. **Regarding the parameter G, we extended our supplementary experiments to larger values (N907MA181M-D10,20,50B-G40).** The results demonstrate strong extrapolation capability, with our formula achieving significantly better performance against baselines in terms of fitting errors (0.0085 vs. 0.0647 vs. 0.0702).
>
> - Robustness across random seeds. To control computational costs, a fixed random seed (seed=1234) was used in most experiments, along with sliding window smoothing. **To assess robustness, we conducted supplementary several experiments with different random seeds (seed=1234, 1122, 3344).** The results demonstrate minimal variance with an average standard deviation of only 0.000849.
>
> We believe these supplementary experiments and clarifications substantially enhance the persuasiveness of our paper and adequately address the reviewers’ concerns. We sincerely appreciate the reviewers’ valuable comments and constructive feedback.
>
> We would like to express our sincere gratitude again for your meticulous review. The valuable comments and insightful suggestions from the reviewers have not only guided our corresponding revisions and improvements but also greatly enriched the depth of our work. We remain committed to further refining and enhancing our research in the future.
>
> Sincerely, The Authors

---

### Meta-Review · Area_Chair_pNqh · 2026-01-11

**Summary:**

While all three reviewers provided borderline positive ratings (6, 6, 6), and I genuinely appreciate the authors' substantial experimental effort, I recommend rejection. I view this work as fundamentally an empirical study that sweeps MoE hyperparameters and fits curves to the results. Although the experiments are comprehensive, it seems that the contribution may not yet be sufficient, as it does not deepen our understanding of why these particular parameter values are optimal. I am also concerned that the findings may lack actionable guidance for practitioners working at production scale, since it remains unclear whether these conclusions generalize beyond the authors' experimental setup. For an empirical scaling law paper to meet ICLR's bar, it should offer either novel theoretical insights or validation at scale, and I hope the authors can strengthen the work in these directions for future submission.

**Reviewer Concerns:**

The authors invested substantial experimental effort in this work, which is appreciated. However, this is fundamentally an empirical study of MoE hyperparameters. Empirical studies of this nature have a high bar at ICLR, and the contribution does not clear that bar in the current form.

Reviewers raised concerns about statistical robustness (no confidence intervals on fitted coefficients), contradiction with OLMoE findings on shared experts, comparison fairness, and confounding factors dismissed as beyond scope. These were partially addressed but not fully resolved.

**Reviewer Scores:**

All three reviewers provided borderline positive ratings of 6. While reviewers acknowledged the substantial experimental effort and recognized some improvements during rebuttal, they did not strongly advocate for acceptance. The uniformly borderline scores reflect appreciation for the effort without strong enthusiasm for the contribution.

---

### Decision · Program_Chairs · 2026-01-26

Reject